# A Reproducibility Study of Counterfactual Explanations for Image Classification

## Abstract

Counterfactual explanations have gained traction in recent years due to their contrastive and potentially actionable nature: moving an outcome from the original class to an alternative target class. Generating plausible and accurate counterfactuals remains challenging. We highlight two underexplored but critical factors influencing counterfactual quality for image classifiers: the neural network architecture and the chosen target class. This work presents a comprehensive empirical evaluation of multiple counterfactual explanation methods across diverse neural architectures and all possible target classes on the MNIST and CIFAR-10 datasets. Our results show that performance can vary substantially across architectures and targets, an aspect often overlooked in prior evaluations. To better assess counterfactual explanation plausibility, we introduce a novel evaluation method based on Moran's index, a spatial autocorrelation metric. This allows us to systematically identify and exclude structurally implausible counterfactuals that existing metrics may overlook. We find that counterfactual explanation methods often fail to generate counterfactual explanations for intended target classes, due to factors such as timeouts, restrictive search spaces, or implementation issues. Furthermore, our analysis demonstrates that evaluating explanations on only one target class or architecture provides an incomplete and potentially misleading picture of performance. Additionally, we show that different plausibility metrics do not consistently agree, emphasising the need for more robust evaluation frameworks. In summary, we (i) identify architecture and target class as key overlooked dimensions in counterfactual explanation performance, (ii) propose a novel plausibility assessment method using Moran's index, and (iii) provide actionable insights for the development and evaluation of more generalisable counterfactual explanation methods.

## 1 Introduction

In recent years, the field of eXplainable Artificial Intelligence (XAI) has gained a lot of traction when it comes to making black-box models understandable (Nauta et al., 2023). Through time, models and tasks have become increasingly complex, making the black-box even more opaque. This results in an a growing need for post-hoc explanations that can accurately describe the inner workings of the models. A popular post-hoc explanation method is a counterfactual explanation, which are based on asking 'what if' questions for model predictions, i.e., 'What if the inputs to my model had been different?'. In particular, when considering counterfactual explanations, we are interested in minimally changing some factors to observe if this small change would have affected the outcome. Whilst counterfactual explanations offer interesting insights into causal relations between the input and output of a model on a per-instance level (Stepin et al., 2021), the generated explanations cannot always be justified (Laugel et al., 2019) and are not necessarily robust to model changes (Jiang et al., 2023). Additionally, prior theoretical work analysing attribution methods, including counterfactual explanations, proved that counterfactuals are always recourse sensitive and thus can never be robust (Fokkema et al., 2023). This inherent lack of robustness in counterfactual explanations exemplifies the need for rigourous evaluation.

A major challenge in XAI is that there is no agreed-upon strategy for evaluating the goodness' of an explanation (Hedström et al., 2023; Nauta et al., 2023). Within the area of counterfactual explanations, there

is also no uniform and generally accepted set of evaluation techniques (Stepin et al., 2021). Benchmarks often have different implementations of the same counterfactual explanation algorithm and of the metrics used to evaluate those explanations (Karimi et al., 2022; Le et al., 2023), making it impossible to compare the performance of counterfactual explanation methods. This can lead to inconsistencies in explanation quality depending on the implementation used. Furthermore, assessing counterfactual explanations based on averages of metrics instead of considering performance gaps between subgroups can unfairly favour one method over another (Balagopalan et al., 2022). The problem with these trends is that they will likely further stifle the evaluation of counterfactual explanation methods and thus their use in practice, as there is no real understanding of how well current approaches perform.

Whilst there have been efforts to standardise counterfactual explanation methods for tabular data (Pawelczyk et al., 2021; Verma et al., 2020), analyse decision boundaries in time series classification (Baer et al., 2025), and there have been efforts to analyse post-hoc explanation methods in general (Bodria et al., 2023; Klein et al., 2024), there has not been the same level of analysis in the case of counterfactual explanations for image classification. A notable exception to this is the work by Velazquez et al. (2023) where a wide variety of counterfactual explanation methods are evaluated. However, the results are based on a single synthetic dataset, they do not include an analysis over a variety of network architectures and they do not analyse the complete scope of possible decision boundaries.

In this work, we address this gap by analysing the current state of counterfactual explanations for image classifiers and identifying key challenges that affect their evaluation and reliability. We make the following contributions:

1. We conduct a systematic empirical evaluation of five counterfactual explanation methods across two image classification datasets (MNIST and CIFAR-10), multiple neural architectures, and all possible target classes – highlighting how architecture and target class significantly affect counterfactual explanation quality.

2. We propose a novel plausibility evaluation method based on the spatial autocorrelation measure known as Moran's I (Moran, 1948), which enables the identification and exclusion of structurally implausible counterfactuals.

3. We show that current methods often fail to generate counterfactuals for intended target classes due to timeouts, search space restrictions, or implementation issues.

4. We demonstrate that existing plausibility metrics frequently disagree, underscoring the need for more robust evaluation tools.

Through our experiments,[1] we reproduce five different implementations of counterfactual explanations from Kenny & Keane (2021) and Van Looveren & Klaise (2021), and evaluate them using a variety of metrics, following a similar approach to that of Klein et al. (2024). While we do not seek to add to the clutter of evaluation measures in XAI, we argue that Moran's I captures unique structural information relevant to plausibility, and can complement existing metrics.

With our experimental design, we aim to answer the following research questions:

1. How do the different counterfactual explanation methods perform across a variety of evaluation metrics?

2. To what extent does the performance of different counterfactual explanation methods change for different neural architectures?

3. To what extent does the performance of different counterfactual explanation methods change for different decision boundaries?

---

[1]Resources to support the reproducibility of this paper are available at https://anonymous.4open.science/r/CFX-benchmarking-33B9

4. To what extent do different implementations of plausibility agree with one another?

Research question 1 covers the performance of counterfactual explanations using the evaluation metrics implemented, and analyses how successful each method is, including reasons for failure. We also showcase the use of our proposed measure. Research questions 2 and 3 examine the impact of architectural and decision boundary variation. Finally, research question 4 focuses on the consistency between different plausibility metrics, including our own.

The paper is organised as follows. Section 2 provides background on counterfactual explanations and their evaluation. Section 3 introduces our proposed evaluation method. Section 4 details the experimental setup, including the counterfactual explanation methods and evaluation metrics. Section 5 presents the results, followed by discussion and conclusions in Section 6.

## 2 Background

### 2.1 Defining counterfactual explanations

To formally define counterfactual explanations, we must first consider the model task that we are explaining. We note that this depends on what we define to be an instance and what is being perturbed. In the case of image classification, the instance is the image that we are classifying in the test set, and the perturbations are individual pixel changes within that instance. As broadly described by Guidotti (2022), given a classifier $f(x)$, where $x$ is the instance we are classifying with decision $y = f(x)$, a counterfactual explanation $x'$ is defined as a perturbed $x$, such that the decision of $f$ has changed $f(x') \neq f(x)$, and that it does so minimally, i.e., we would like to minimise $\mathrm{dist}(x', x)$, where $\mathrm{dist}(\cdot, \cdot)$ is some distance function.

A broader definition of counterfactual explanations accounts for multiple examples of $x'$ being generated, that is, a counterfactual explainer could generate the set $C = \{x'_1, x'_2, \ldots, x'_h\}$. These examples could encompass different aspects of a single decision and rely on inherently different input values.

Other definitions of counterfactual explanations could include desirable properties that the explainer should optimise for. For example, plausibility concerns a counterfactual explanation that is realistic to the original dataset distribution. Another way to describe this is as justified counterfactual explanations (Laugel et al., 2019), that is there should be a continuous path between the counterfactual explanation and an instance from the training data, using the concept of an $\epsilon$-chain to ensure this property holds. More desirable properties that are especially of interest to this work are validity (satisfying $f(x') \neq f(x)$), minimality (there should not exist a valid counterfactual example $x''$ with a smaller distance to $x$) and similarity (the explanation $x'$ should be similar to $x$, i.e. $\mathrm{dist}(x', x) < \epsilon$).

### 2.2 Counterfactual explanations for image classifiers

Since counterfactual explanations were initially described by Wachter et al. (2017) in a loan application setting, many adaptations have been developed to fit different data types better. Initial applications of counterfactual explanations in images proposed a minimum edit solution that would greedily optimise towards a target image with the intended target class (Goyal et al., 2019). However, unlike in the domain of tabular data, where a minimum edit approach can work well, in images, the potential number of pixel changes is too large to generate an interpretable explanation. Thus, to ensure that counterfactual explanations remain true to the original data distribution, many approaches have turned to generative models such as Generative Adversarial Networks (GANs) and Variational Autoencoders (VAE) to be sure that the explanations lie close to the original image in the latent space (Guidotti et al., 2019; Joshi et al., 2019; Kenny & Keane, 2021; Liu et al., 2019). Other methods approach the problem slightly differently, such as using prototypes to guide their counterfactual explanations (Van Looveren & Klaise, 2021), using reinforcement learning (Samoilescu et al., 2023) and energy-based modelling (Altmeyer et al., 2024).

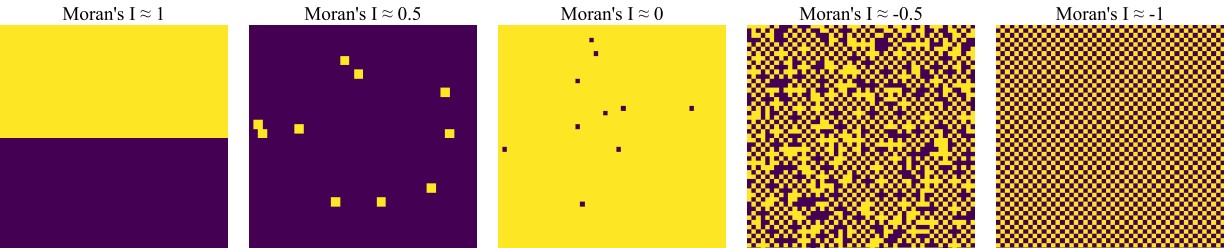

Figure 1: Illustration of spatial patterns corresponding to different values of Moran's I. From left to right, the images depict decreasing spatial autocorrelation: 1.0 (perfect positive autocorrelation, highly clustered), 0.5 (moderate clustering), 0 (random spatial pattern, no autocorrelation), –0.5 (moderate negative autocorrelation, alternating patterns), and –1.0 (perfect negative autocorrelation, complete spatial dispersion).

### 2.3 Evaluation metrics

The evaluation of counterfactual explanations is still an open problem in the field. Whilst there are a lot of different methods to evaluate counterfactual explanations for different ideal properties, the challenge is still to accurately capture these properties within a metric. Nauta et al. (2023) provide an overview of the metrics used in XAI research along with the corresponding property it is trying to capture, known as the Co-12 properties. For example, several measures try to capture the concept of faithfulness or correctness, meaning how faithful the explanation is to how the model works. One approach to capturing this is the faithfulness correlation metric introduced by Bhatt et al. (2021), which is implemented for any feature-based explanation. Other metrics for feature-based explanations include incrementally flipping pixels or changing each feature to measure the effect it has on the model (Arya et al., 2019; Bach et al., 2015; Montavon et al., 2018). The main problem in capturing the faithfulness of counterfactual explanations is the lack of a ground truth explanation to compare the output explanation to. A possible solution to this is to generate a synthetic dataset where the ground truth explanation can be controlled (Liu et al., 2021; Oramas et al., 2019).

Another commonly tested measure is that of plausibility, which aims to capture how interpretable the generated explanation is and whether it is close to the original data distribution. When proposing the CEM method, Dhurandhar et al. (2018) train an autoencoder on the training dataset to ensure that the generated explanations are close to the data manifold. A similar approach is taken by Van Looveren & Klaise (2021) with the IM1 and IM2 metrics, where an autoencoder is trained on each class. Alternatively, a measure for plausibility suggested by Kenny & Keane (2021) uses Monte Carlo dropout, which is commonly used for out-of-distribution detection. A caveat of this approach is that it makes use of the dropout of a neural network, which means the network that needs to be explained must have some dropout in its layers for this metric to be used. The measure R% substitutability is introduced as a way to measure how well counterfactual explanations can substitute the training data (Samangouei et al., 2018). Lastly, Altmeyer et al. (2024) modify the plausibility metric proposed by Guidotti (2022), which is the distance of the counterfactual explanation from its nearest neighbour in the target class.

## 3 Moran's Index as a Diagnostic Measure

As mentioned previously, there are many plausibility measures, which mostly try to capture how close a counterfactual explanation is to the original data distribution. Whilst this approach is conceptually sound, these methods often require generative models, and they may be hard to interpret. Thus, we propose using a simple statistic known as *Moran's I*. Global Moran's I is known as a measure of spatial autocorrelation and is often used in geographical analysis (Moran, 1948). The values can range from $-1$ to 1. A simple illustrative example is provided in Figure 1 that visually shows these possible values. A Moran's I of 1 represents high spatial correlation, and a value close to 0 represents randomness. On the other hand, a Moran's I value of $-1$ indicates a perfect separation between each pixel. The index is generally defined as

$$I = \frac{n}{W} \cdot \frac{\sum_{i=1}^{n} \sum_{j=1}^{n} w_{ij}(x_i - \bar{x})(x_j - \bar{x})}{\sum_{i=1}^{n}(x_i - \bar{x})^2}, \tag{1}$$

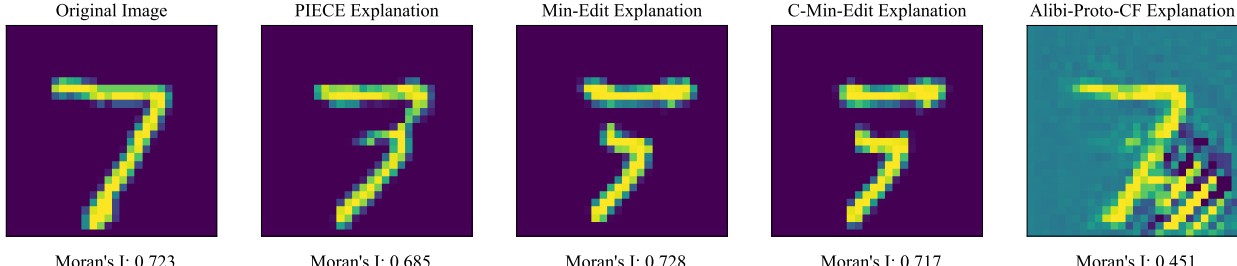

| Original Image | PIECE Explanation | Min-Edit Explanation | C-Min-Edit Explanation | Alibi-Proto-CF Explanation |
|---|---|---|---|---|
| Moran's I: 0.723 | Moran's I: 0.685 | Moran's I: 0.728 | Moran's I: 0.717 | Moran's I: 0.451 |

Figure 2: Original image with different counterfactual explanations produced by different methods, along with their MI scores. The original model prediction was a "7", and the target is a "3". In this case, the right most explanation has a lower MI score than the other explanations, mainly due to the checkerboard pattern in the bottom right corner.

where $n$ is the number of spatial units, in the case of images, these would be pixels. Then $x_i$ is the value of the variable of interest at location $i$ and $\bar{x}$ is the mean of the variable $x$ over all locations. Lastly, $w_{ij}$ is the spatial weight between location $i$ and location $j$ (typically from a chosen spatial weights matrix) and $W$ is the sum of all spatial weights, $W = \sum_{i=1}^{n} \sum_{j=1}^{n} w_{ij}$. Note that the spatial weight matrix decides which neighbouring features are important for $x_i$ and thus what is chosen can depend on what information is important in $x_i$. We choose to use the queen kernel, which counts all directions as possible neighbours. For example, in a $3 \times 3$ kernel, all 8 cells surrounding the cell of interest are considered neighbours and would be weighted with 1.

We assume that higher spatial autocorrelation relates to better interpretability of explanations, as illustrated in Figure 2. Thus, we propose using Moran's I (MI) as a diagnostic measure for explanation quality in images, as we believe it can filter out certain types of implausible explanations. We envision two ways of using it for explanations. The first is to simply compute the MI value of each explanation, where higher values are better. An alternative is to compute the absolute difference between the MI of the original image and the MI of the explanation. The reasoning for this is that the MI score might be more meaningful if we look at it relative to the original image. This means that lower values would be considered better.

# 4 Experimental Design

In this section, we describe the design of our experiments. First, we outline the networks that were trained and tested. Next, we detail the counterfactual explanation methods applied to interpret these models. Finally, we explain the evaluation metrics and procedures used to assess both model performance and the quality of the explanations.

## 4.1 Networks

In order to answer the research questions listed in Section 1, we train multiple architectures on two datasets. As our first dataset, we use MNIST (LeCun, 1998) as it is the only dataset on which most, if not all, counterfactual generation methods have been evaluated. The second dataset we choose to train the architecture on is CIFAR-10 (Krizhevsky, 2009).

The architectures used are summarised in Table 1 (see Appendix A for implementation details). As some counterfactual explanation methods are written in PyTorch and others in Keras, we needed to translate the same neural architecture across these two different frameworks. For *mnist_mlp_relu_4_1024* we used an ONNX model to store the network weights, which could then be converted into either a Keras or Torch model using the learned weights. This led to many complications in aligning the weights to the network architecture in the corresponding package. For example, the method alibi-CF failed to produce counterfactual explanations for almost every instance and decision boundary, but this only occurred with the ONNX converted model. In every other case, we fully trained both Keras and PyTorch models separately, ensuring that the translated architecture was correct. Still, there were differences in test accuracy, the largest difference being 4.3% and 5.3% on the CNN and ResNet8 networks for CIFAR, respectively. For the other networks,

Table 1: Neural architectures and test performance on both Keras and PyTorch for the MNIST and CIFAR-10 datasets.

| Architecture | Torch Accuracy | Keras Accuracy | Dataset |
|---|---|---|---|
| MLP Relu_4_1024 | 0.9848 | 0.9848 | MNIST |
| CNN (Kenny & Keane, 2021) | 0.9925 | 0.9934 | MNIST |
| CNN (Van Looveren & Klaise, 2021) | 0.9720 | 0.9819 | MNIST |
| Lenet5 (LeCun et al., 1998) | 0.9726 | 0.9882 | MNIST |
| ResNet8 | 0.9918 | 0.9924 | MNIST |
| MLP (Schut et al., 2021) | 0.9476 | 0.9659 | MNIST |
| CNN in (Kokhlikyan et al., 2020) | 0.5779 | 0.6208 | CIFAR-10 |
| ResNet8 | 0.7724 | 0.7196 | CIFAR-10 |
| ResNet18 (He et al., 2016) | 0.8387 | 0.8314 | CIFAR-10 |

this difference is either roughly equal to or less than 1%. Note that this means the underlying models were not necessarily perfectly the same for both Keras and Torch.

## 4.2 Counterfactual explanation methods

We reproduce the Alibi-based counterfactual explanations (Klaise et al., 2021; Van Looveren & Klaise, 2021) and the PIECE method (Kenny & Keane, 2021) and its corresponding baselines. We chose to reproduce these methods because they represent different ways of generating counterfactual explanations. The main differentiating factor between all these methods is the loss function that is used to optimise towards a counterfactual. The most straightforward approach is the alibi-CF method, which is roughly based on the work of Wachter et al. (2017) and does not use any generative model to guide the explanation. Following this, the Min-Edit and C-Min-Edit both use a GAN $G$ and a latent representation of the original image $z$ in their optimisation step to get to the latent representation of the explanation $z'$. The main difference between these two is how they define the loss. The PIECE method also uses this same $z$ and $G$, but adds an additional factor using feature values in the original image that are less likely to occur in the target class, named exceptional features. Lastly, the alibi-Proto-CF method uses an autoencoder to add a prototype for each target class. A corresponding loss term is added to guide the explanation towards the target class and speed up the search process. We provide an overview of the optimisation step of each of the counterfactual explanation methods in Table 2.

Table 2: Overview of the optimisation formulations and Python frameworks used for each counterfactual explanation method that was tested in our experiments.

| Method | Optimisation step | Python library |
|---|---|---|
| **alibi-CF** | $L(x' \mid x) = \big(f_t(x') - p_t\big)^2 + \lambda L_1(x', x)$ | Keras |
| **Min-Edit** | $z' = \arg\min_z \|f(c(G(z))) - Y_{c'}\|_2^2$ | PyTorch |
| **C-Min-Edit** | $z' = \arg\min_z \max_\lambda \lambda \|f(c(G(z))) - Y_{c'}\|_2^2 + \mathrm{dist}(c(G(z)), x)$ | PyTorch |
| **PIECE** | $z' = \arg\min_z \|c(G(z)) - x'\|_2^2$ | PyTorch |
| **alibi-Proto-CF** | $L = sL_{\mathrm{pred}} + \beta L_1 + L_2 + L_{AE} + L_{\mathrm{proto}}$ | Keras |

Here $p_t$ denotes the target class probability and $f_t(\cdot)$ is the predicted class probability. The term $\lambda$ is a parameter that balances the trade-off between prediction accuracy and feature value similarity. The function $c(\cdot)$ denotes all layers of the classifier up until the penultimate layer and $f(\cdot)$ is the output of the classifier itself. The term $Y_{c'}$ represents the output of the classifier after the Softmax operation, yielding a probability vector corresponding to the target class $c'$. Lastly, for alibi-Proto-CF, $L_{\text{pred}}$ corresponds to the model's prediction function with scaling parameter $s$. The next two terms represent the elastic net regularisation, with parameter $\beta$, and $L_{AE}$ and $L_{\text{proto}}$ representing the autoencoder and prototype losses. There is a subtle distinction between the Alibi-based methods and the remaining approaches: in the latter, $z'$ denotes a latent vector, whereas in the Alibi methods, $L$ more generally represents a loss function.

These counterfactual explanation methods were chosen because Alibi is a better-known toolkit, often used as a baseline and should thus provide well-maintained functions. We include PIECE as we aim to reproduce it and is a method focused on generating plausible counterfactual explanations. We generate counterfactual explanations for 100 instances on each potential target class.

### 4.3 Evaluation metrics

We evaluate the counterfactual explanation methods using metrics for correctness, plausibility, size, and running time, which aligns with prior work on evaluating counterfactual explanations (Guidotti, 2022). We focus on the following properties: correctness, success, size, running time and plausibility. The first metric is simply the time it takes to optimise a counterfactual explanation, referred to as **OT** in this paper. Then we are interested in computing the size of a counterfactual explanation. This is done using the $L_1$, $L_2$ and $L_\infty$ -norm between the original and the perturbed instance. Lastly, three proxy measures for plausibility are implemented, including our implementation of *MI*.

- **Validity/Correctness**: Does the new prediction of the counterfactual explanation change to the target class (excluding incorrect timeouts)?

- **Success**: How many counterfactual explanations can the method successfully produce (including timeouts)?

- **Success Ratio**: What fraction of the source-target pairs per image successfully produce a counterfactual explanation (including timeouts)?

- **Optimisation Time**: Time it took to run the method.

- **Size/Minimality**: $L_1$, $L_2$, and $L_\infty$ distance between $x$ and $x'$.

- **IM1/IM2** (Van Looveren & Klaise, 2021): The IM1 and IM2 metrics are based on the reconstruction loss of autoencoders trained on each class:

$$\text{IM1}(x', y, y') = \frac{\|x' - AE_y(x')\|_2^2}{\|x' - AE_y(x')\|_2^2 + \epsilon} \tag{2}$$

$$\text{IM2}(x', y') = \frac{\|AE_y(x') - AE(x')\|_2^2}{\|x'\|_1 + \epsilon}, \tag{3}$$

  where $AE_y$ represents the output vector of an autoencoder trained on a particular class $y$. We note that, based on prior research, IM2 is not necessarily interpretable (Mahajan et al., 2019) and has been shown not to provide significantly different outcomes on out-of-distribution data (Schut et al., 2021).

- **(Im)-plausibility** (Altmeyer et al., 2024): Based on a definition by Guidotti (2022), implausibility is computed:

$$\text{impl}(x', X_{y_+}) = \frac{1}{|X_{y_+}|} \sum_{x \in X_{y_+}} \text{dist}(x', x) \tag{4}$$

  where $X_{y_+}$ is a subsample of the training dataset for the class $y_+$. The distance function can be chosen; however, for images, the structural similarity method (Wang et al., 2003) provides more insights into image similarity than the Euclidean norm would.

- **MI**: Moran's I of the explanation $MI(x')$. Alternatively we compute $|MI(x) - MI(x')|$ or $\Delta$ MI as a potential variation of this metric.

We make a distinction between the *correctness* and the *success* of an explanation because an explanation is not necessarily incorrect, but just unsuccessful in finding a solution within the time limit. Had the explanation method been given more time, it may have produced something correct. Thus, we consider success to be a fairer evaluation of correctness.

### 4.4 Summary

Summarising, we run experiments with a total of 90 different experimental settings, i.e., 9 architectures, 2 datasets, and 5 counterfactual explanation methods, that we assess using 9 evaluation metrics.

## 5 Experimental Results

Our experimental results are structured according to our four research questions in Section 1. We first evaluate counterfactual explanation performance overall. Then, we study the importance of network architecture and target classes. Lastly, we investigate correlations between plausibility metrics.

### 5.1 RQ1: Evaluating the performance of counterfactual explanation methods

The results addressing the performance of various counterfactual explanation methods across multiple evaluation metrics are presented in Table 3. Starting with the measure of validity, the success ratio is highest overall for C-Min-Edit. It is important to note, however, that the table does not distinguish between success and correctness, particularly in scenarios involving incorrect instances that timeout. Timed out instances are where the method has exhausted the maximum number of iterations. After excluding cases in which the counterfactual explanations either timed out, produced incorrect results, or both, both alibi methods consistently yielded correct counterfactual explanations, with one notable exception. Specifically, nine instances of incorrect counterfactual explanations occur with the alibi-CF method on the MLP_relu_1024 network for the MNIST dataset. In these cases, the produced counterfactual explanations all predicted the same class as the original image class label.

This leads to the question: Why do counterfactual explanation methods fail to produce correct results? While we can only hypothesise, three primary reasons likely contribute. First, the method may simply time out, as each approach operates under a predefined iteration limit. Second, suboptimal hyperparameter settings might overly constrain the search space, preventing successful outcomes. This is particularly evident in the C-Min-Edit, Min-Edit, and PIECE methods, where continuous increases in loss suggest the algorithm is stuck, potentially requiring either additional time or a less restricted search space. Finally, there is always the possibility of bugs in the original source code, which can result in incorrect counterfactual explanations before reaching a timeout.

Moving on to other aspects of Table 3, the smallest counterfactual explanations are generated by alibi-CF for the MNIST dataset and by PIECE for the CIFAR dataset. Regarding optimisation time, there appears to be minimal difference between the overall optimisation time for all counterfactuals and that for the correct ones. Notably, the increased complexity of the CIFAR dataset significantly extends the optimisation time for the alibi-based methods, whereas the other methods demonstrate improved efficiency in their running times. This observation is somewhat unexpected, as it might be assumed that dealing with RGB values would generally pose greater challenges for counterfactual explanation methods in identifying optimal solutions. A potential explanation could be the shared common denominator among these methods: they all utilise the same GAN architecture and latent representation of the original image to generate their counterfactual explanations. If this GAN is working better on CIFAR than on MNIST, this may impact how the methods perform on a different dataset. Regarding the three different plausibility measures implemented, PIECE demonstrates the best $MI(x')$ and implausibility scores across both datasets, with Min-Edit performing comparably as a close second. For the IM1 measure, alibi-Proto-CF appears to yield the most favourable results. Beyond these observations, the agreement between the various plausibility measures remains ambiguous. Looking

Table 3: Average performance of the different investigated counterfactual explanation methods according to different performance metrics for all architectures for MNIST and CIFAR datasets. The best-performing method according to each metric for both datasets is indicated in bold print. Values are reported as mean ± standard deviation in parentheses. ST stands for success ratio and OT for optimisation time. Both *ST all* and *OT all* are calculated over all counterfactual explanations, while all other metrics are only evaluated over correct counterfactual explanations, i.e., counterfactual explanations that change the prediction of the model to the predefined target class. Impl. refers to the implausibility metric and MI to Moran's I index. Both the average of the difference in MI metric between the original image and the counterfactual explanation and the average of the MI metric for the counterfactual explanation are included. Arrows indicate whether a higher or lower value for the metric is better, using ↑ and ↓ respectively.

| Method | ST all ↑ | OT all ↓ | OT ↓ | $L_2$ ↓ | IM1 ↓ | IM2 ↓ | Impl. ↑ | $\Delta$ MI ↓ | MI(x') ↑ |
|---|---|---|---|---|---|---|---|---|---|
| **MNIST** | | | | | | | | | |
| C-Min-Edit | **0.69** (±**.27**) | 33.82 (±26.05) | 20.67 (±16.63) | 15.70 (±3.83) | 3.31 (±9.82) | 0.1087 (±.23) | 0.56 (±.06) | 0.08 (±.08) | 0.69 (±.11) |
| Min-Edit | 0.62 (±.28) | **19.43** (±**15.78**) | **9.63** (±**9.85**) | 15.84 (±3.79) | 3.46 (±10.35) | 0.1079 (±.23) | 0.57 (±.06) | 0.08 (±.08) | 0.70 (±.11) |
| PIECE | 0.27 (±.24) | 28.48 (±13.86) | 10.62 (±11.19) | 15.49 (±4.03) | 3.03 (±10.57) | **0.0875** (±**.20**) | **0.57** (±**.06**) | 0.08 (±.08) | **0.70** (±**.11**) |
| alibi-CF | 0.61 (±.40) | 54.69 (±41.97) | 66.24 (±45.20) | **5.42** (±**1.84**) | 1.60 (±2.71) | 0.1106 (±.20) | 0.50 (±.09) | **0.06** (±**.04**) | 0.68 (±.09) |
| alibi-Proto-CF | 0.67 (±.29) | 111.39 (±58.34) | 96.01 (±47.39) | 23.71 (±8.31) | **1.08** (±**.73**) | 0.2532 (±.16) | 0.14 (±.15) | 0.25 (±.23) | 0.50 (±.23) |
| **CIFAR** | | | | | | | | | |
| C-Min-Edit | **0.99** (±**.03**) | 10.62 (±7.60) | 10.47 (±7.06) | 16.12 (±5.77) | 1.05 (±.24) | 0.0017 (±0.0008) | 0.0099 (±.02) | 0.05 (±.04) | 0.86 (±.08) |
| Min-Edit | 0.98 (±.06) | **6.83** (±**6.93**) | **6.42** (±**5.98**) | 16.13 (±5.77) | 1.05 (±.24) | 0.0017 (±0.0008) | 0.0101 (±.02) | 0.05 (±.04) | 0.86 (±.08) |
| PIECE | 0.75 (±.29) | 16.85 (±14.59) | 10.10 (±9.07) | **15.31** (±**4.66**) | 1.04 (±.24) | 0.0016 (±0.0007) | **0.0105** (±**.02**) | 0.05 (±.04) | **0.87** (±**.07**) |
| alibi-CF | 0.92 (±.18) | 107.99 (±77.33) | 112.16 (±78.70) | 32.12 (±6.41) | 1.08 (±.28) | **0.0008** (±**0.0003**) | -0.0058 (±.03) | **0.01** (±**.04**) | 0.82 (±.09) |
| alibi-Proto-CF | 0.66 (±.18) | 298.58 (±101.32) | 287.69 (±91.23) | 36.70 (±8.97) | **1.02** (±**.16**) | 0.0040 (±0.0040) | -0.0070 (±.02) | 0.35 (±.29) | 0.50 (±.31) |

at our measure MI, the simpler formulation $MI(x')$ proves to be more informative than $|MI(x) - MI(x')|$. This is because the latter formulation creates substantially smaller variations across methods, making it less effective for distinguishing their performance. This is why we see alibi-CF have a high $MI(x) - MI(x')$ and the smallest size in MNIST, because the explanations are likely very close to the original image and thus would have a very similar *MI* score to the original. However, this does not necessarily mean that it is more plausible, as explanations can change the decision to the target but still look roughly the same. Interestingly, the method with the lowest $MI(x')$ overall is alibi-Proto-CF, which may relate to our anecdotal evidence in Figure 2.

Bringing these points together to answer RQ1; it seems there is no clear winner across metrics. However, across both datasets the best performing method for each metric is mostly the same. The key insight is that the effectiveness of explanation methods varies depending on which value is prioritised—be it size, optimisation time, or plausibility—as no single method excels across all criteria.

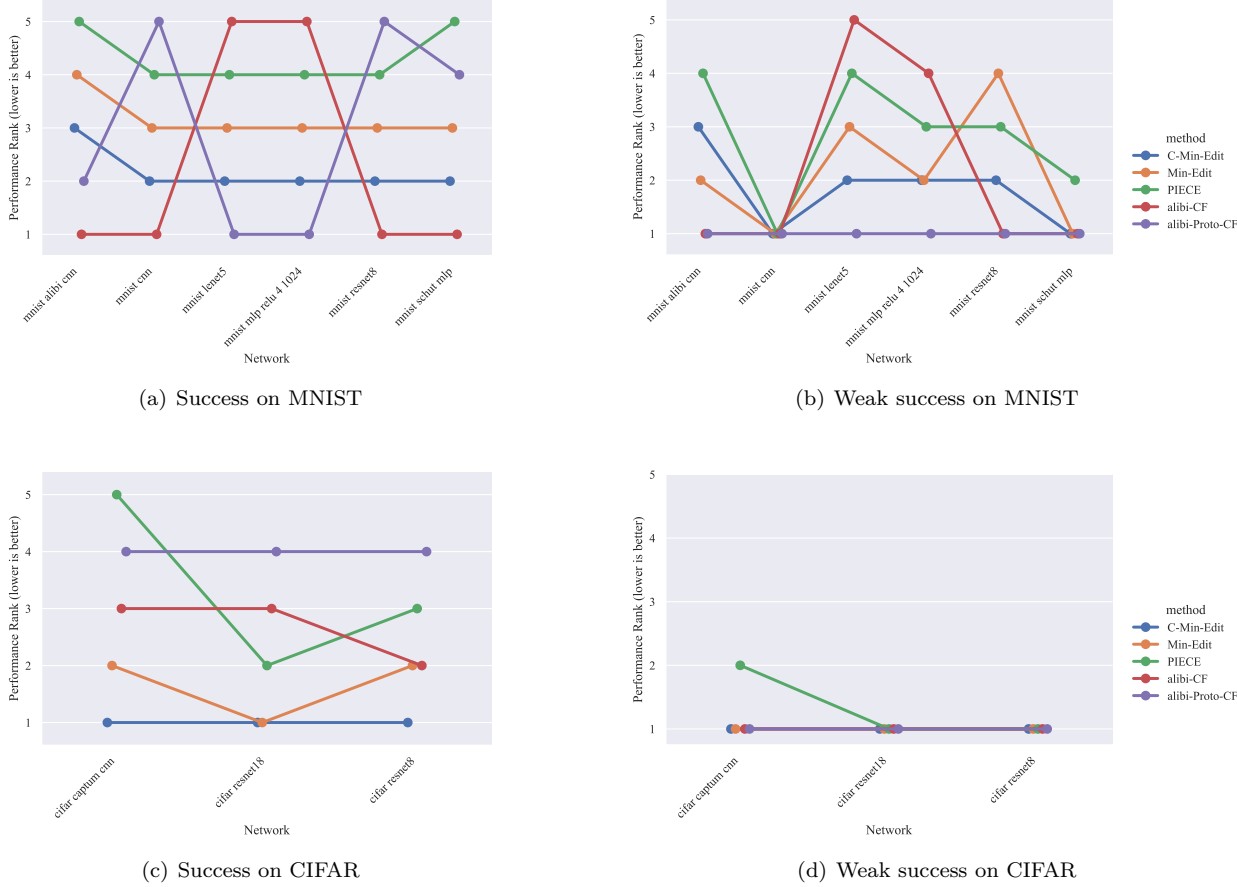

(a) Success on MNIST

(b) Weak success on MNIST

(c) Success on CIFAR

(d) Weak success on CIFAR

Figure 3: Performance Rank of the average success and weak-success rates for each counterfactual explanation method across all MNIST networks. 'Success' and 'weak success' are binary indicators denoting whether a generated counterfactual explanation fully or partially achieves the target class, respectively. For each network, the average success rate per method is computed, and methods are ranked accordingly; lower ranks indicate better performance.

## 5.2 RQ2: Network architectures

While it may seem intuitive, the performance of a counterfactual explanation method can vary significantly depending on the network architecture employed. In our analysis, we observed substantial variation between networks. For instance, when comparing the MLP Relu_4_1024 to the CNN utilised in (Kenny & Keane, 2021), alibi-Proto-CF achieves near-perfect performance on the former, but its success rate drops to 27.7% on the latter. To further illustrate this across all networks, Figure 3 presents an overview of all evaluated networks for both datasets, ranking methods on success and weak success metrics. The results highlight a clear distinction between different networks: whilst alibi-Proto-CF ranks highest on the networks MLP_Relu_4_1024 and LenNet5, it also ranks worst on ResNet8 and the CNN tested in (Kenny & Keane, 2021). In particular, we observe that for the success metric, the relative ranking of methods can change substantially depending on the network. This indicates that a method that performs well on one network may perform relatively poorly on another. This effect is less pronounced when evaluated with the weak success metric. Notably, alibi-Proto-CF consistently identifies at least one explanation across all networks. For CIFAR, we again observe evidence of reordering under the success metric, though to a lesser extent than in MNIST, while performance under the weak success metric remains largely unchanged across methods and networks.

To summarise, we found that at least for the success or correctness rate of a counterfactual explanation, network architecture impacts explanation method performance. In particular, we found that two methods

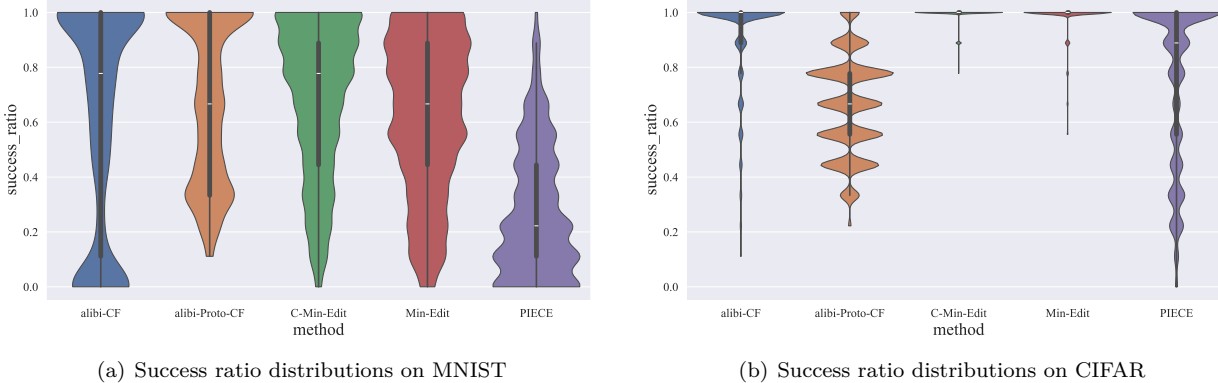

(a) Success ratio distributions on MNIST          (b) Success ratio distributions on CIFAR

Figure 4: Success ratios for each counterfactual explanation method across MNIST and CIFAR datasets.

can have complimentary success on two different neural networks. This implies that evaluating explanations on one network may result in misleading findings.

### 5.3 RQ3: Decision boundaries

Another aspect of this research is to determine to what extent different decision boundaries can impact the performance of an explanation method. To show an overview of this impact, Figure 4 shows for each method the distribution of the success ratio, with lower values indicating that a lower number of decision boundaries were successfully crossed. Here, it is clear that for the method PIECE and to some lesser degree the others, a lot of instances are concentrated around the lower success rates. This indicates that not all decision boundaries can be successfully reached, and methods like alibi-Proto-CF are better at finding more of these decision boundaries than a method like PIECE.

For the more challenging CIFAR dataset, the success ratio distributions appear considerably more irregular and dispersed across methods. Compared to MNIST, the distributions tend to be wider around a success ratio of 1, while exhibiting very thin tails across most methods. This means that a success ratio less than 1 is much less common for CIFAR than it was for MNIST. Notably, alibi-Proto-CF displays an unusual oscillating, wave-like pattern, a feature that is also visible, though to a lesser extent, for the PIECE method.

We also compare the percentages of how many explanations are successful and weakly successful per method. Weakly successful allows for the evaluation to be more lenient, stating that one decision boundary per image is enough to be successful. We see a big jump in success for all methods in Table 4. Once we make a weaker assumption and accept any one decision boundary to suffice, we see all methods essentially producing successful counterfactual explanations for every image.

Additional evidence on specific original-to-target class transitions and their corresponding decision boundaries is presented in Figure 5 for the PIECE method. The figure illustrates that certain class pairs are considerably easier to shift between. For example, moving an image from a 5 to a 6 or from a 4 to a 9 scores relatively well, whereas moving from a 0 to a 7 is far less successful. This observation aligns with intuition: some transitions are inherently more plausible given the visual and semantic similarities between classes. Corresponding heatmaps for the remaining methods are provided in Appendix B.

Beyond addressing this research question, these results also highlight the influence of excluding unsuccessful timeouts. As shown in Figure 5(b), removing timeouts leads to a consistent increase in average success across all decision boundaries. This suggests that a substantial portion of failures arises from methods simply exhausting their allotted optimisation time. However, since success values do not reach 1 across all class pairs, it is evident that additional factors — such as an overly constrained search space or possible code bugs — also contribute to the generation of unsuccessful counterfactual explanations.

Table 4: A comparison of success and weak success as a percentage over the different methods across MNIST and CIFAR datasets.

| Method | Success (%) | Weak Success (%) |
|---|---|---|
| **MNIST** | | |
| C-Min-Edit | 68.65 | 98.82 |
| Min-Edit | 61.69 | 97.64 |
| PIECE | 27.22 | 78.79 |
| alibi-CF | 60.55 | 76.60 |
| alibi-Proto-CF | 67.21 | 100.00 |
| **CIFAR** | | |
| C-Min-Edit | 99.30 | 100.00 |
| Min-Edit | 98.48 | 100.00 |
| PIECE | 75.48 | 99.00 |
| alibi-CF | 91.74 | 100.00 |
| alibi-Proto-CF | 65.78 | 100.00 |

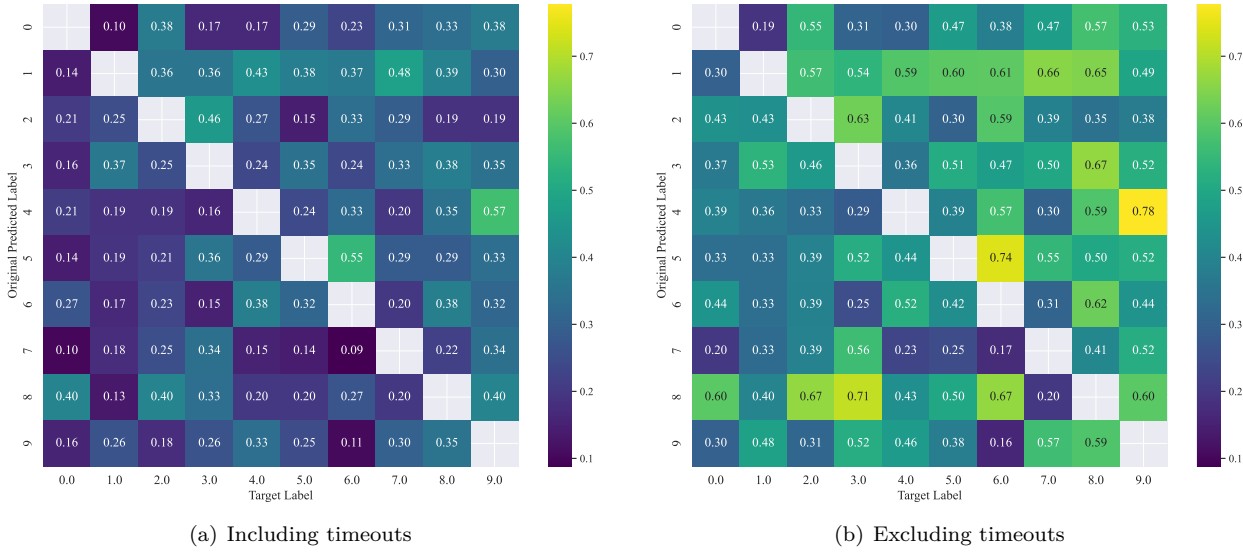

(a) Including timeouts          (b) Excluding timeouts

Figure 5: Average Success for each original class and target class pair for the PIECE method on MNIST. Timeouts are either included in the average success calculation or filtered out.

Taken together, our findings indicate that the choice of target decision boundary significantly influences the performance of counterfactual explanation methods. This observation is consistent with theoretical expectations, as counterfactuals function by introducing minimal perturbations to the input. When the target decision boundary is situated too far from the original instance, identifying a viable counterfactual within a constrained optimisation timeframe becomes increasingly infeasible. We find that some methods are better at finding these less feasible decision boundaries than others. Thus, we advise practitioners to take this into consideration when evaluating counterfactual explanations.

### 5.4 RQ4: Plausibility metric agreement

The final research question addresses whether the various plausibility measures exhibit agreement with one another. Although all measures aim to assess the same underlying concept of plausibility, they are computed through fundamentally different approaches. As illustrated in Figure 6, there is no consistent alignment

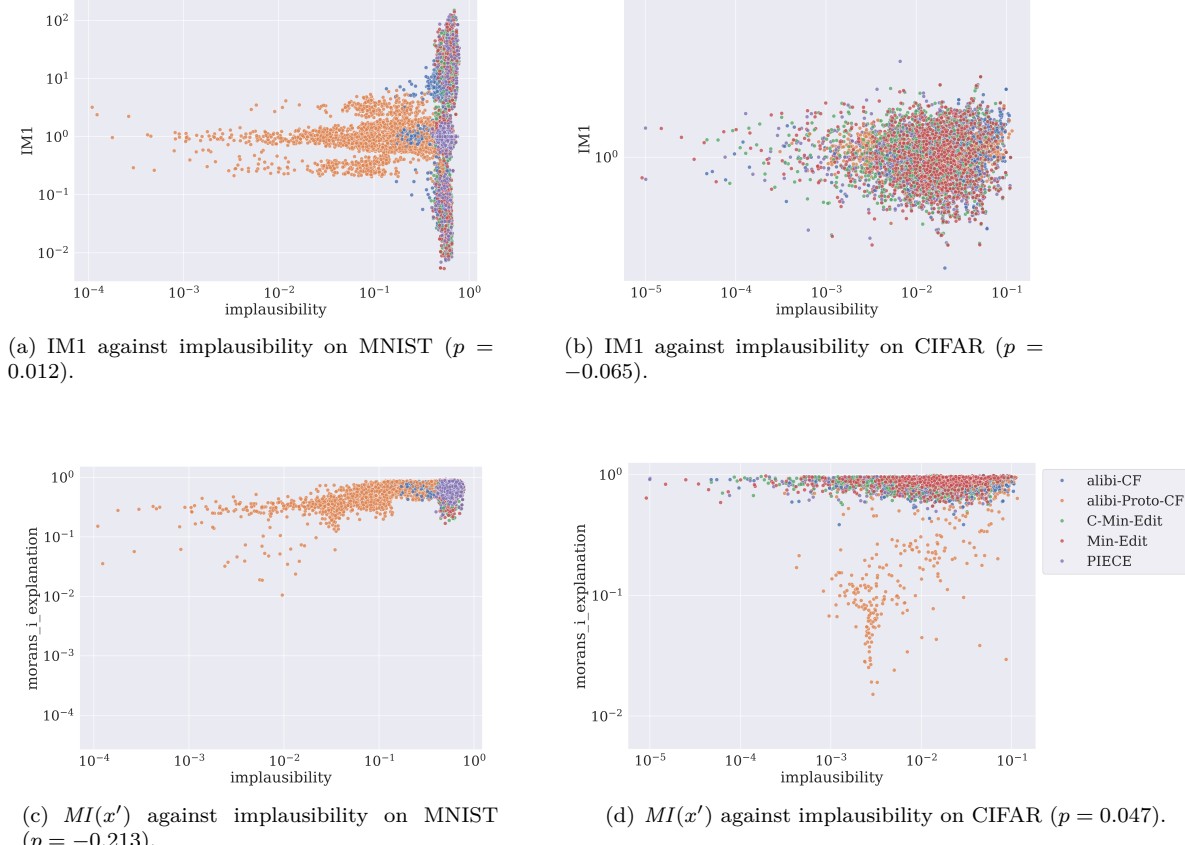

(a) IM1 against implausibility on MNIST ($p = 0.012$).

(b) IM1 against implausibility on CIFAR ($p = -0.065$).

(c) $MI(x')$ against implausibility on MNIST ($p = -0.213$).

(d) $MI(x')$ against implausibility on CIFAR ($p = 0.047$).

Figure 6: Scatter plots showing agreement between plausibility measures for different explainability methods, colored by method. We test whether the measures are correlated using a Pearson correlation test. The corresponding p-values are shown in the subfigure captions.

between the metrics. For instance, in subfigure (a), a distinct cluster is observed for the alibi-Proto-CF method, differentiating it from the other methods; however, aside from this, no clear patterns emerge. A similar observation can be made in subfigure (b), where no discernible distinctions between the methods are evident. The results presented in Table 3, which displays the average performance of each method across the metrics, suggest some potential agreement between implausibility and $MI(x')$. However, we do not observe any clear linear relationships; instead, both cases of subfigures (c) and (d) showcase a distinct cluster of points associated with alibi-Proto-CF. This suggests that the explanations generated by the alibi-Proto-CF method tend to be more distinctly implausible in comparison to those produced by the other methods.

Overall, we find that there is mostly no agreement between the tested plausibility measures. We attribute this to the different ways in which they are computed, thus capturing different aspects of plausibility. Based on this, we argue that relying exclusively on a single plausibility metric is insufficient to provide practitioners with definitive guidance on the plausibility of explanations. This further supports the notion that human validation may be essential for reliably assessing the plausibility of counterfactual explanations (Keane et al., 2021).

## 6 Conclusion

In this study, we have provided an extensive analysis of the performance of counterfactual explanation methods in the domain of image classification. We specifically focused on their performance, their interaction with

network architectures, the influence of decision boundaries, and the agreement between different plausibility measures.

**Main findings.**   Our results reveal several key insights. First, in terms of performance, methods like C-Min-Edit and PIECE demonstrate strong results in generating valid counterfactual explanations. However, when examining plausibility measures, no clear alignment between the different measures emerge, suggesting that different metrics capture different aspects of plausibility. Notably, whilst our proposed spatial autocorrelation measure, Moran's I, show similar results on average to the implausibility measure, when investigating further, we see that this agreement might be limited to the prototype-based method alibi-Proto-CF. In other words, the consistency between the two measures does not seem to generalize across other counterfactual generation methods. Regarding network architectures, our findings indicate that the performance of counterfactual explanation methods can significantly vary depending on the architecture used. Similarly, the chosen target class also plays a critical role in determining success rates, with some methods being more adept at navigating these boundaries than others.

**Implications.**   One of the important broader implications of our work is that to properly assess the effectiveness of counterfactual explanation methods, one should ensure that explanations can be generated for a variety of decision boundaries and do so on different neural architectures. By empirically evaluating counterfactual explanations across different experimental settings we both address the broader calls for addressing evaluation in explanations and we provide a first step towards a benchmark for practitioners.

**Limitations.**   Whilst the research conducted in this paper is extensive, with 90 experimental settings, there are some limitations. To be more comprehensive, there could have been a larger variety of explanation methods implemented, a deeper analysis of their hyperparameters and a larger variety of evaluation methods tested, particularly for the faithfulness metric. Furthermore, expanding the experiments to include additional datasets such as MedMNIST (Yang et al., 2023), would strengthen the generalisability of the findings.

**Future work.**   As to future work, we want to generalise our findings in multiple directions. First, we plan to evaluate counterfactual explanation methods in other settings than image classification, such as in recommender systems (Baklanov et al., 2025; Mohammadi et al., 2025). Second, we plan to conduct a user study to evaluate the effectiveness of the produced counterfactual explanations and measure the agreement of the user evaluation with the plausibility metrics that we tested. Finally, we view this empirical investigation as a foundational step toward a more rigorous theoretical analysis of counterfactual explanations. In particular, the emerging field of formal explainable AI, which seeks to establish formal guarantees for the properties of counterfactual explanations, represents a promising direction for future research.

## Resources

To facilitate reproducibility of the results and findings of this paper, we share the following resources at https://anonymous.4open.science/r/CFX-benchmarking-33B9:

- Notebooks for creating the visualizations found in the paper.

- Code for generating counterfactual explanations, including the latent representations and pickle files needed for the PIECE method.

- Code for training the networks that were used in this paper.

- Results with computed metrics in the form of cleaned csv files.

- Raw results of each counterfactual explanation method and all networks.

## Acknowledgements

Hidden to preserve anonymity.

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

## Appendix

## A    Experimental Details

### A.1    Network architectures

### MLP Relu_4_1024

This network was the only network that we did not train. It was sourced from a neural network verification tool, Eran.[2] The original format was a PyTorch model. This was then translated using the ONNX file format into a keras-tensorflow model. The specifics of the neural architecture can be found in Table 5.

| Layer | Details |
|---|---|
| Input | $1 \times 28 \times 28$, flattened to $1 \times 784$ |
| Hidden Layer 1 | Fully Connected, 1024 units |
| ReLU | |
| Hidden Layer 2 | Fully Connected, 1024 units |
| ReLU | |
| Hidden Layer 3 | Fully Connected, 1024 units |
| ReLU | |
| Output Layer | Fully Connected, 10 units |

Table 5: Neural network architecture for `mnist_relu_4_1024`, also named `mnist_output_100` in the results.

### mnist_cnn

This network was used by Kenny & Keane (2021). We chose to use the exact same specifications as the original. The only change is that we reduced the number of epochs from 3 times 10 epochs at different learning rates, to just 10 epochs. We use a learning rate of 0.001 and a batch size of 8.

### mnist_alibi_cnn

This network was used in the notebook showcasing the method by Van Looveren & Klaise (2021) implemented in the alibi package.[3] The hyperparameters used were the same as in the alibi notebook. We used 3 epochs, a batch size of 32 and a learning rate of 0.001.

### LeNet5

The LeNet5 network (LeCun et al., 1998) was used. The number of epochs was 10, the batch size 8 and the learning rate 0.001.

### ResNet8

We chose to use a shallow ResNet architecture as an alternative to the other networks because larger ResNets would have been too much for MNIST. The hyperparameters that we used were 10 epochs, a batch size of 8 and a learning rate of 0.001.

We used the same neural architecture for CIFAR and changed the channels to work for RGB data. However, with the number of epochs increasing to 30, a batch size of 32 and a learning rate of 0.001.

### MLP

This network is based on the network used by Schut et al. (2021). We do not use an ensemble as in the original paper. We use a batch size of 8, a learning rate of 0.001 and 20 epochs.

---

[2]https://github.com/eth-sri/eran
[3]https://docs.seldon.io/projects/alibi/en/stable/examples/cfproto_mnist.html

| Layer | Details |
|---|---|
| Conv2d | 8 filters, $5 \times 5$, stride $1 \times 1$, padding $= 2$ |
| Dropout2d | $p = 0.1$ |
| BatchNorm2d | |
| ReLU | |
| Conv2d | 16 filters, $5 \times 5$, stride $2 \times 2$, padding $= 2$ |
| Dropout2d | $p = 0.1$ |
| BatchNorm2d | |
| ReLU | |
| Conv2d | 32 filters, $5 \times 5$, stride $1 \times 1$, padding $= 2$ |
| Dropout2d | $p = 0.1$ |
| BatchNorm2d | |
| ReLU | |
| Conv2d | 64 filters, $5 \times 5$, stride $2 \times 2$, padding $= 2$ |
| Dropout2d | $p = 0.2$ |
| BatchNorm2d | |
| ReLU | |
| Conv2d | 128 filters, $3 \times 3$, stride $1 \times 1$, padding $= 1$ |
| GAP | Global Average Pooling |
| Linear | |
| SoftMax | 128 input units, 10 output units |

Table 6: Summary of the CNN architecture in (Kenny & Keane, 2021), also referred to in the results as `mnist_cnn_output_100`.

| Layer | Details |
|---|---|
| Input | $1 \times 28 \times 28$ image |
| Conv2d | $1 \rightarrow 64$, kernel $2 \times 2$, padding $= 1$ |
| ReLU | |
| MaxPool2d | kernel $2 \times 2$, stride $2 \times 2$ |
| Dropout | $p = 0.3$ |
| Flatten | $64 \times 14 \times 14 \rightarrow 12544$ |
| Fully Connected (fc1) | $12544 \rightarrow 256$ |
| ReLU | |
| Dropout | $p = 0.5$ |
| Fully Connected (fc2) | $256 \rightarrow 10$ |

Table 7: Summary of the `alibi_CNN` architecture.

**cifar_captum_cnn**

The network architecture used in the documentation of Kokhlikyan et al. (2020).[4] The batch size is 32, the number of epochs is 30 and the learning rate is 0.001.

**ResNet18**

We use a slightly modified version of He et al. (2016). We made the network shallower to allow for easier training. We used the code of the accompanying GitHub repository and removed the fourth block.[5] The batch size chosen was 32, the learning rate 0.001 and the number of epochs 60.

---

[4] https://github.com/pytorch/captum/blob/09aa0489b19b09f5fa9238dffedd1a79d277b620/tutorials/CIFAR_Captum_Robustness.ipynb#L542
[5] https://github.com/huyvnphan/PyTorch_CIFAR10/blob/master/cifar10_models/resnet.py

| Layer | Details |
|---|---|
| Input | $1 \times 28 \times 28$, flattened to $1 \times 784$ |
| Hidden Layer 1 | Fully Connected, 80 units |
| BatchNorm | |
| ReLU | |
| Hidden Layer 2 | Fully Connected, 80 units |
| BatchNorm | |
| ReLU | |
| Output Layer | Fully Connected, 10 units |

Table 8: Summary of architecture for `MLP`, also known in the results file as `mnist_schut_mlp`.

| Layer | Details |
|---|---|
| Input | $3 \times 32 \times 32$ |
| Convolution Layer 1 | 6 filters, $5 \times 5$ kernel, stride $= 1$ |
| ReLU | |
| Max Pooling Layer 1 | $2 \times 2$ kernel, stride $= 2$ |
| Convolution Layer 2 | 16 filters, $5 \times 5$ kernel, stride $= 1$ |
| ReLU | |
| Max Pooling Layer 2 | $2 \times 2$ kernel, stride $= 2$ |
| Fully Connected Layer 1 | 120 units |
| ReLU | |
| Fully Connected Layer 2 | 84 units |
| ReLU | |
| Output Layer | Fully Connected, 10 units |

Table 9: Summary of architecture for `cifar_captum_cnn`.

### A.2 Explanation method specifications

**PIECE and baselines**

All hyperparameters of these methods were kept the same as in the original paper. The only exception is that we increased the number of runs from 300 to 1000 for PIECE, and also set the number of runs to 1000 for the baselines.

**alibi-CF**

We used the documentation of alibi[6] and set the following hyperparameters:

- gamma $= 100$

- theta $= 100$

- c_init $= 1$

- c_steps $= 2$

- $k = 5$

- max_iterations $= 1000$.

---

[6]https://docs.seldon.io/projects/alibi/en/stable/examples/cf_mnist.html

**alibi-Proto-CF**

We used the documentation of alibi (see footnote 6) and set the following hyperparameters:

- target_proba = 1.0

- tol = 0.1

- max_iter = 1000

- lam_init = 1e-1

- max_lam_steps = 10

- learning_rate_init = 0.1.

### A.3   Evaluation metric specifications

Specifically for the IM1 and IM2 metrics, we used the same autoencoder architecture as implemented in (Kenny & Keane, 2021), for both MNIST and CIFAR. For the Moran's I measure, we use what is known as a queen kernel. That is, every neighbouring pixel (left, right, diagonal) is processed.

### A.4   Hardware specifications

To generate all explanations we used an Intel Xeon Gold 5118 CPU with 48 logical processors and a base frequency of 2.30 GHz. All other operations were conducted on a Lenovo Yoga Slim 7 Pro; Processor AMD Ryzen 7 5800H; Memory 16 GB DDR4 3200 MHz; Storage 512 GB SSD; AMD Radeon Graphics (CPU used only).

## B   Additional Visualisations

In this section, we present additional visualisations that complement and reinforce the findings discussed in the main body of the paper. Specifically, Figures 7 to 15 illustrate results analogous to those shown in Figure 5, but extended across all evaluated methods and for both the MNIST and CIFAR datasets.

Across nearly all methods and datasets, a consistent improvement in the average success rate is observed once timeout cases are excluded. This pattern provides additional evidence that timeouts contribute significantly to failures in generating valid counterfactual explanations, particularly in regions of the input space where decision boundaries are more complex or computationally demanding to traverse. The main exceptions to this trend appear in Figures 12 and 13, corresponding to the Min-Edit and C-Min-Edit methods, respectively, where the removal of timeouts results in negligible changes. This stability could be attributed to the already high baseline success rates of these methods, with their lowest observed average values being 0.89 for Min-Edit and 0.94 for C-Min-Edit.

A further noteworthy pattern emerges in Figures 10 and 15, which depict results for the alibi-Proto-CF method. These heatmaps reveal distinct vertical structures, indicating that certain target classes consistently yield higher success rates regardless of the original predicted class. This suggests that some target classes are inherently easier to reach or better represented within the prototype space, highlighting potential asymmetries in how counterfactual explanations are distributed across the target domain.

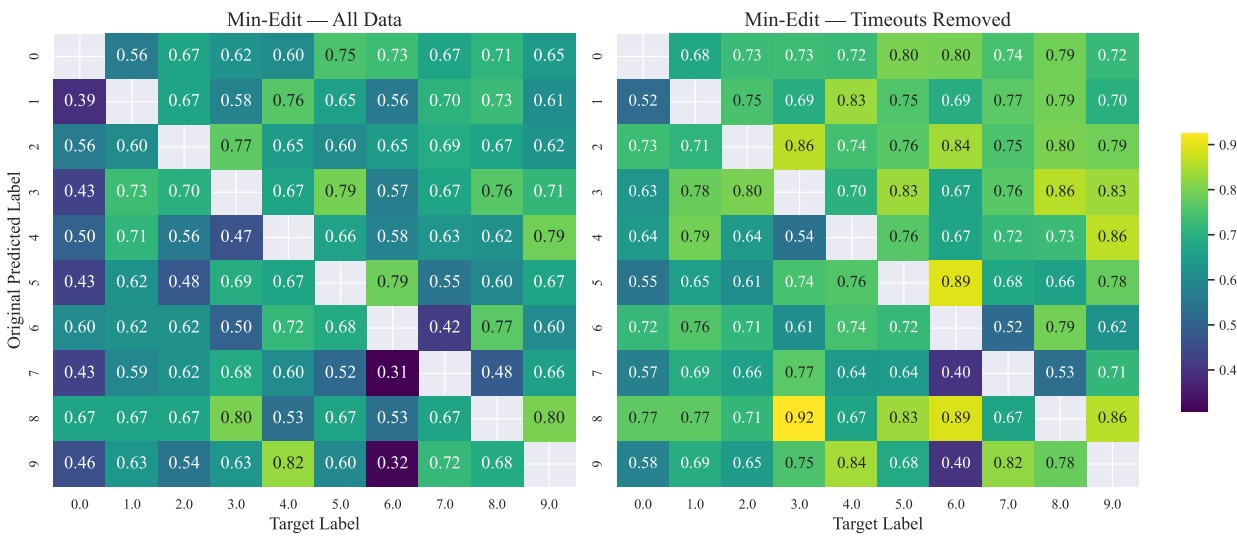

Figure 7: Average Success for each original class and target class pair for the Min-Edit method on MNIST. Timeouts are either included in the average success calculation or filtered out.

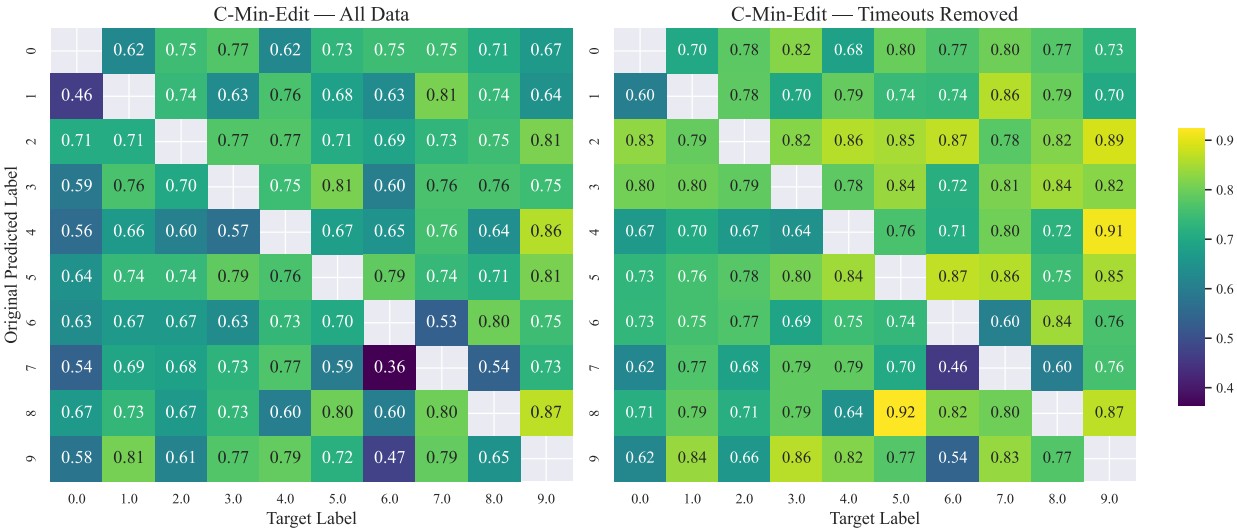

Figure 8: Average Success for each original class and target class pair for the C-Min-Edit method on MNIST. Timeouts are either included in the average success calculation or filtered out.

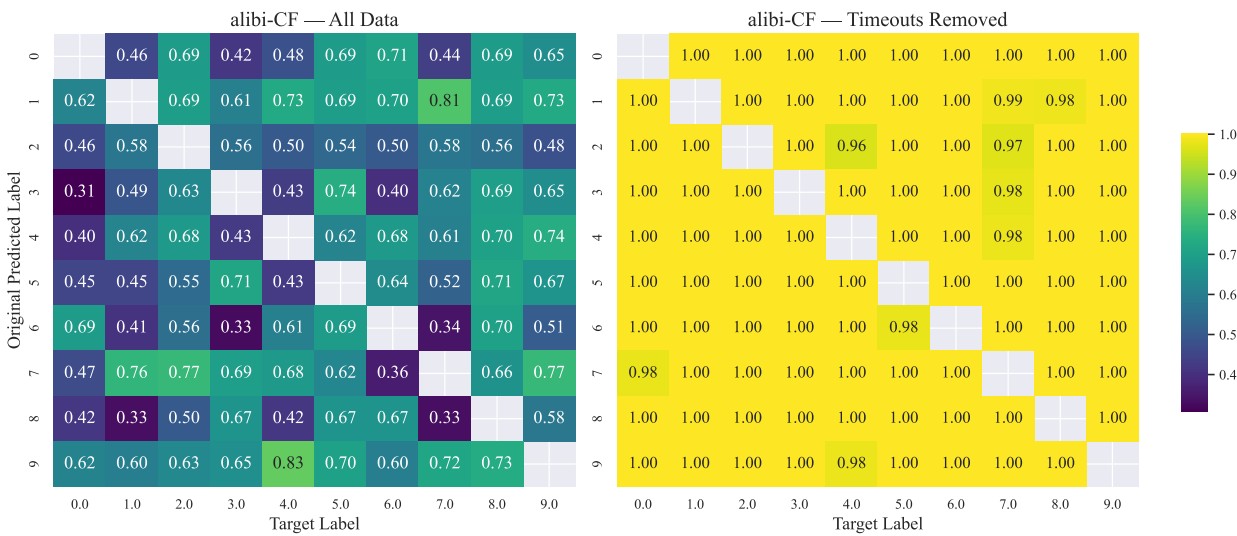

Figure 9: Average Success for each original class and target class pair for the alibi-CF method on MNIST. Timeouts are either included in the average success calculation or filtered out.

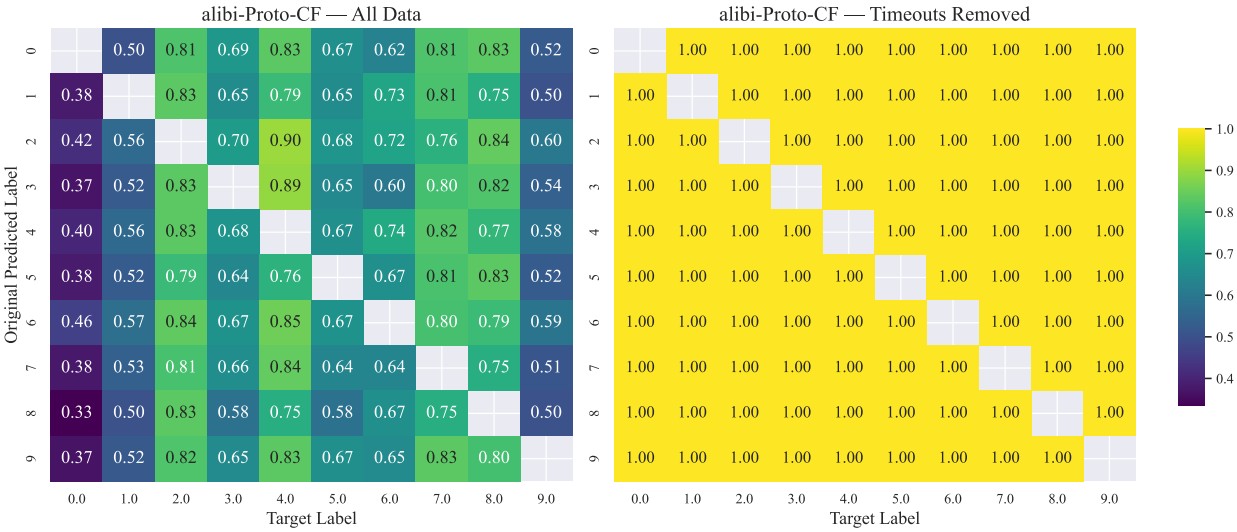

Figure 10: Average Success for each original class and target class pair for the alibi-Proto-CF method on MNIST. Timeouts are either included in the average success calculation or filtered out.

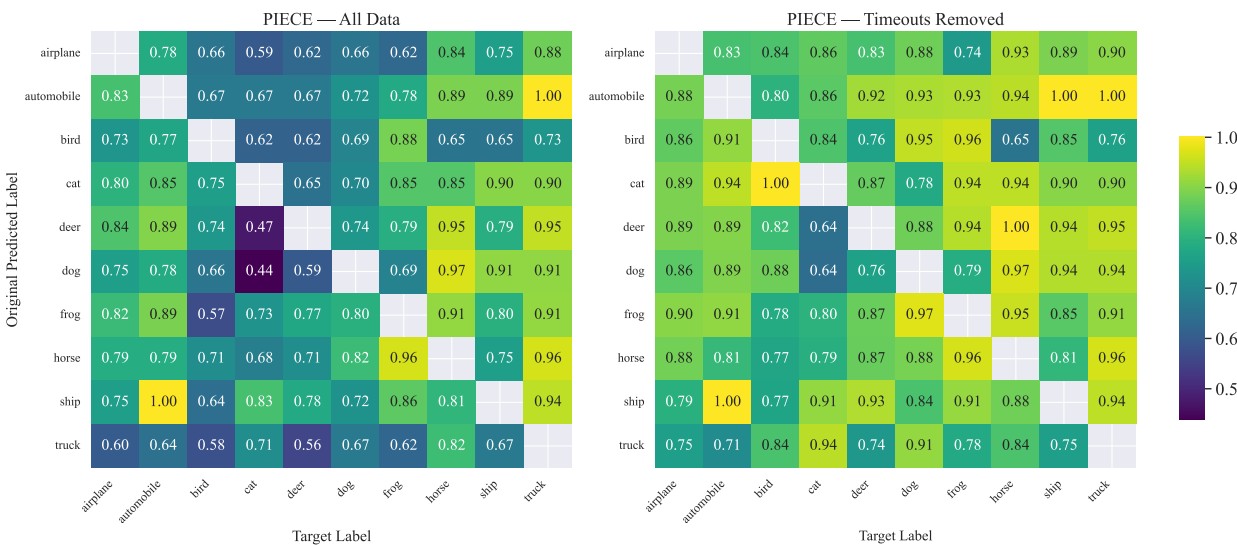

Figure 11: Average Success for each original class and target class pair for the PIECE method on CIFAR. Timeouts are either included in the average success calculation or filtered out.

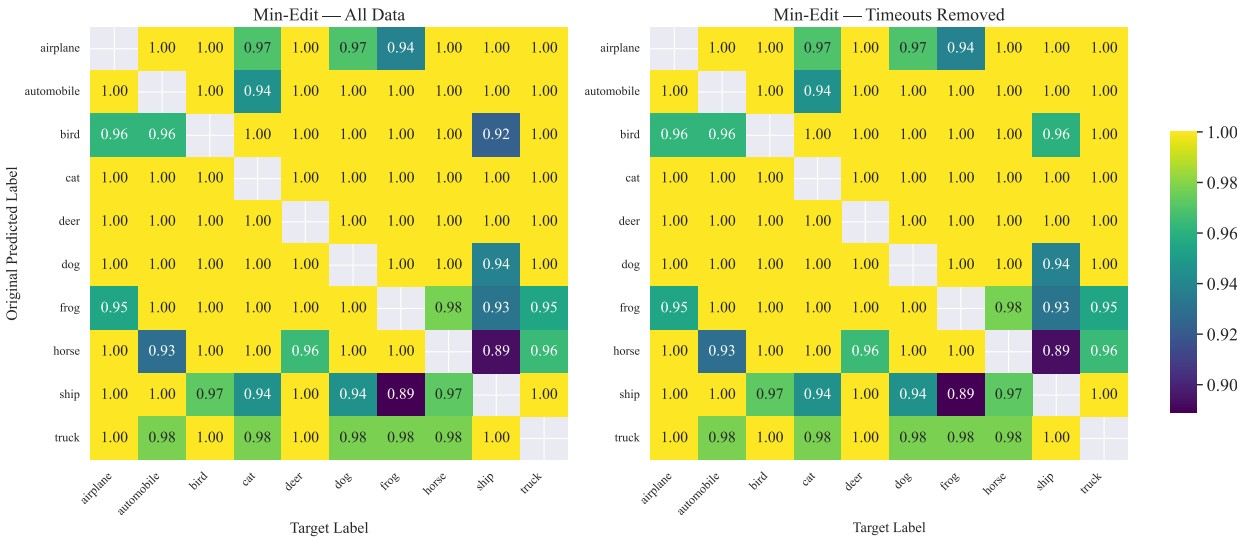

Figure 12: Average Success for each original class and target class pair for the Min-Edit method on CIFAR. Timeouts are either included in the average success calculation or filtered out.

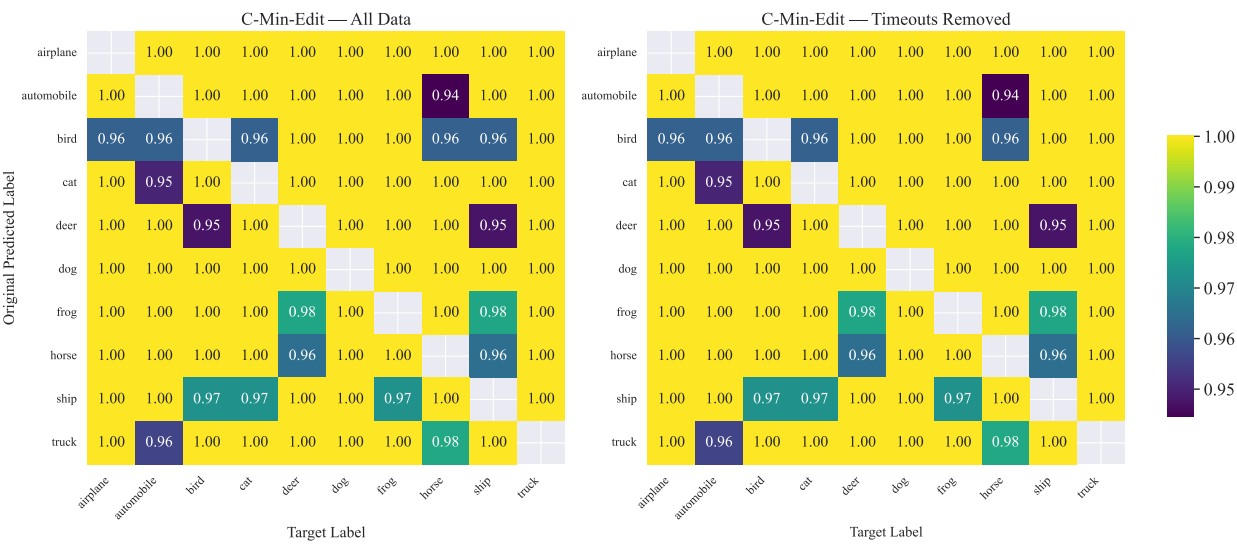

Figure 13: Average Success for each original class and target class pair for the C-Min-Edit method on CIFAR. Timeouts are either included in the average success calculation or filtered out.

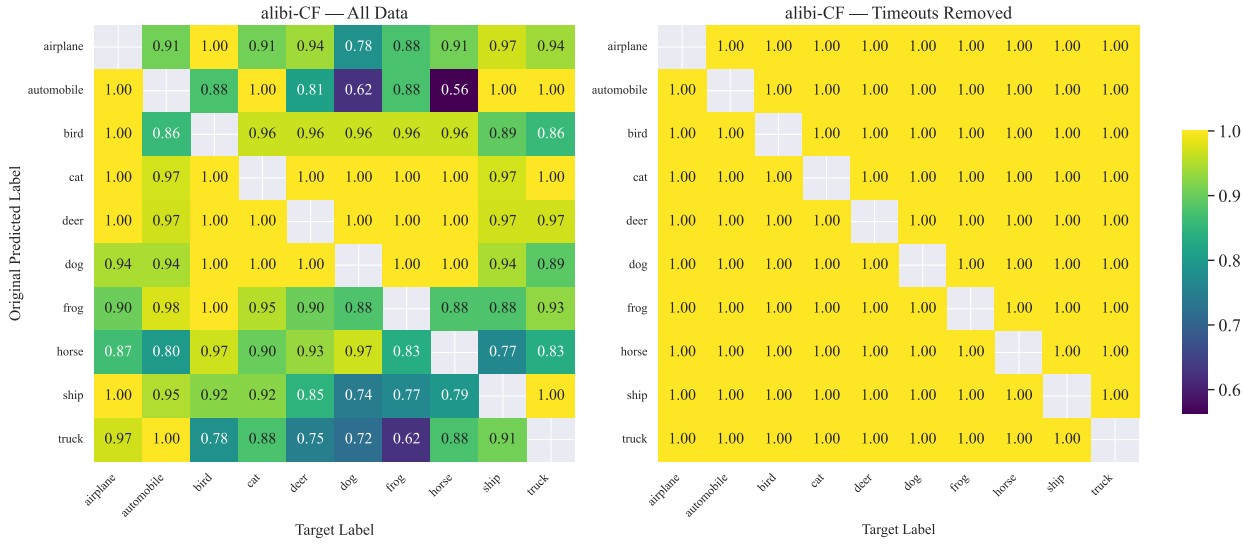

Figure 14: Average Success for each original class and target class pair for the alibi-CF method on CIFAR. Timeouts are either included in the average success calculation or filtered out.

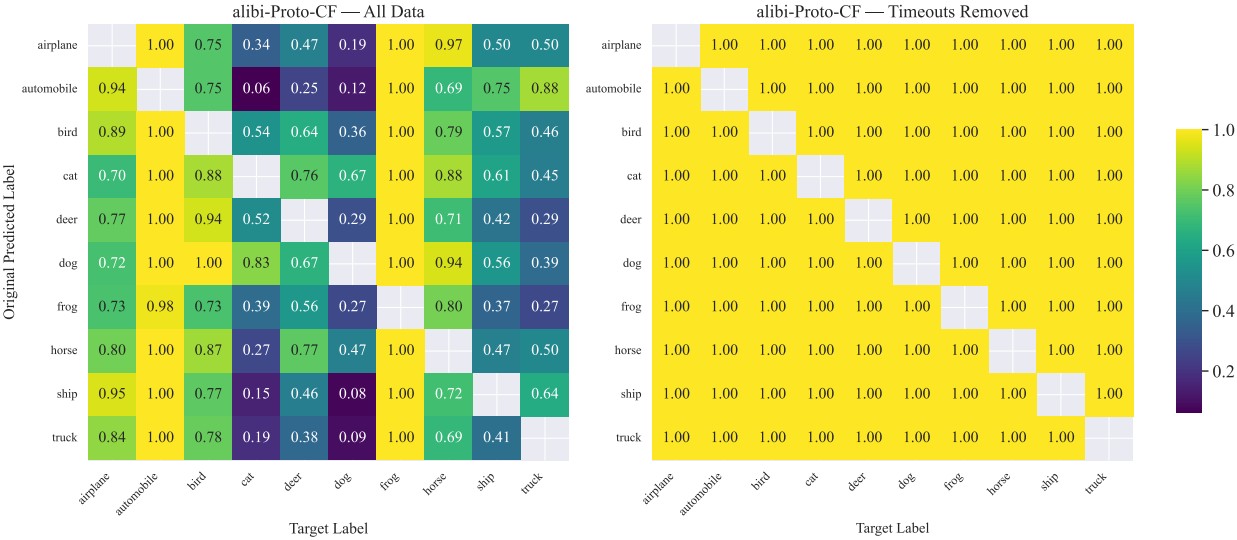

Figure 15: Average Success for each original class and target class pair for the alibi-Proto-CF method on CIFAR. Timeouts are either included in the average success calculation or filtered out.

