# OpenReview forum: "A Reproducibility Study of Counterfactual Explanations for Image Classification"
_TMLR — Rejected by TMLR_

### Review · Reviewer_BesK · 2025-11-21

**Summary Of Contributions:**

This paper presents a reproducibility study and empirical analysis of counterfactual explanation methods for image classification. The key claimed contributions are:

- Architecture and Target Sensitivity: The study establishes that explanation performance varies substantially based on the neural network architecture and the specific source and target classes.

- Moran's I as an Explanation Quality Metric: The authors propose using Moran's I, a spatial autocorrelation metric, as a diagnostic tool to quantify structural plausibility and filter out implausible, high-frequency artifacts (e.g., checkerboard patterns) in generated images.

- Failure Mode Analysis: The analysis reveals that method failures are frequently caused by timeouts and restrictive search spaces rather than model inability; removing timeouts significantly increases success rates across most methods.

- Plausibility Metric Disagreement: The work demonstrates that existing plausibility metrics (IM1, Implausibility) and the proposed Moran's I do not consistently agree, indicating that current metrics capture disjoint aspects of explanation quality.

---

**Strengths**:

1. This paper provides benchmarks, code and metrics to reproduce and compare counterfactual explanations for computer vision classifiers.

2. The paper provides a takeaway, that "to properly assess the effectiveness of counterfactual explanation methods, one should ensure that explanations can be generated for a variety of decision boundaries and do so on different neural architectures". This would demonstrate whether the method generalizes across architectures.

**Weaknesses**:

1. The usage of Moran's I as a metric for counterfactual explanation quality is unjustified. While I agree that it can sometimes be used to flag spurious artefacts in some images, it is unclear how this can be used as a plausibility metric. Does a high Moran's I imply that the resulting image is necessarily a natural image? Or, does a low Moran's I imply necessarily that it is NOT a natural image? Such questions can help answer whether the usage of Moran's I is justified to identify how close counterfactual explanations are to natural images.

2. Evaluation on toy models and datasets: One of the paper's takeaways is that to properly assess effectiveness, explanations must be generated across a variety of settings. However, the paper performs its comparisons primarily on small models, with resnet18 being the largest model, and on toy computer vision datasets, i.e., MNIST and CIFAR-10. It is unclear how effective counterfactual explanations are to more modern settings with larger models (e.g.: vision transformers, convnext models, etc) and higher resolution images (e.g.: imagenet). Note that such a comparison is not hard to do, as no models need to be trained from scratch for this purpose.

3. The majority of the paper's analysis considers "success" metrics, i.e., whether or not a valid counterfactual was generated, but not on any of the interpretability aspects. While plausibility does indeed touch on the quality of the explanations, but then again, as mentioned in Weakness #1, the proposed metric is not well justified. Notably missing is some discussion of downstream utility, or human interpretability. For example, what insights do different methods yield regarding the underlying models or datasets? Do humans find the resulting explanations somewhat interpretable? None of these fundamental questions are addressed by this benchmarking effort.

4. The introduction of the paper mentions that one of the main issue with counterfactuals is robustness, and the paper claims that "this inherent lack of robustness exemplifies the need for rigorous evaluation". However the paper does not attempt to measure the robustness of counterfactuals in their benchmarking.

5. The paper also misses discussions of the links between adversarial examples and counterfactual explanations. For example, see Pawelczyk et al., (2021) Exploring Counterfactual Explanations Through the Lens of Adversarial Examples: A Theoretical and Empirical Analysis; who note that the underlying algorithms are somewhat identical. Further, the robustness literature, it has been found that robust models naturally lead to "perceptually aligned" adversarial examples, or in this case, counterfactual explanations. See, Kaur et al., (2019) Are Perceptually-Aligned Gradients a General Property of Robust Classifiers?, and Santurkar et al., (2019) Image Synthesis with a Single (Robust) Classifier.

**Audience:**

No

**Audience Explanation:**

I would answer no to this question, for the following reasons which have also been elaborated in the weaknesses section:

- The paper does not provide sufficient insights into existing algorithms enough to be interesting, in my view. The major insight is that success criteria might differ across architectures and target classes, which I do not believe will be interesting to the counterfactual explanation researcher.

- The paper does not discuss or consider the downstream utility of such counterfactual explanations, especially for the image domain. The primary pre-requisite for broader interest for interpretability might be some insight into model behavior, or debugging datasets, etc. Unfortunately this paper does not contain such discussion, making it unlikely to be of broader interest.

- The paper is not likely to be of interest to vision researchers. The paper studies counterfactual explanations with small toy datasets (MNIST and CIFAR-10), and small model architectures, which diverges significantly from the norms in computer vision today.

**Broader Impact Concerns:**

There are no broader impact concerns.

**Claims And Evidence:**

No

**Claims Explanation:**

The following claims in the paper may not be accurate:

- One claim is that Moran's I can be used as a measure of explanation quality in images. As discussed in Weakness #1, this claim is not substantiated as a measure of quality or plausibility.

- Another claim is that he performance of a counterfactual explanation method can vary significantly
depending on the network architecture employed. However the paper does not measure this directly, it instead ranks the performs of difference models. This is confusing. Why not directly present success metrics themselves directly? Small deviations in numbers can alter the rank, but do not imply that performance is different.

**Requested Changes:**

I recommend the authors consider the following changes to strengthen the work:

1. First, and foremost would be to perform evaluations on more modern settings when it comes to vision (e.g.: ViTs / ConvNexts on ImageNet) to evaluate whether counterfactual explanations are truly effective for modern computer vision classifiers. This is essential for a benchmark aiming to evaluate whether counterfactual explanations apply to vision.

2. Second, would be to present quantitative justification of Moran's I measure for measuring explanation quality. Why is Moran's I a good measure of explanation quality for natural image data?

3. Third, would be human studies or metrics measuring any downstream utility of such explanations. This must address the question: what concrete benefits do counterfactual explanations have to offer? Example of benefits include: debugging spurious correlations, or being able to answer, "what features does the classifier consider to classify <cats>?", etc.

4. Finally, the authors can consider running their analysis with and without robustness training, to evaluate the benefit of robustness training to counterfactual examples. As discussed in the weaknesses #5, existing work finds robust training to help generate perceptually aligned adversarial examples, which must carry over to counterfactual examples.

---

> ### Author Response · Authors · 2025-12-21
> **Response to Reviewer BesK**
>
> We thank the reviewer for their detailed and constructive feedback and for acknowledging our contributions. We appreciate the time taken to evaluate our reproducibility study of *“On Generating Plausible Counterfactual and Semi-Factual Explanations for Deep Learning”* by Kenny and Keane. Below, we address each concern and outline concrete revisions.
>
> __Weakness #1 and Weakness #3 – On Moran’s I justification and Human Evaluation__
>
> While we justify our use of Moran’s I in Figures 1–2 and Section 3, we agree this justification can be strengthened, particularly through human evaluation. Our intention is not to claim Moran’s I as the best plausibility metric, but as a diagnostic tool for identifying structurally implausible explanations that several methods generate. As shown in Section 5.4 (Figure 6), plausibility metrics correlate weakly, indicating Moran’s I captures a different property.
>
> Regarding the question of whether Moran’s I implies naturalness: a higher Moran’s I reflects more spatially clustered, less spurious pixel changes, which we argue corresponds to greater structural plausibility. Conversely, low Moran’s I indicates noisy, spatially incoherent explanations, which is informative for plausibility assessment.
>
> In response to Weakness #3 (“focus on success metrics rather than interpretability”), we emphasised success rates because they are the most reliable quantitative measure, particularly when targeting distant classes. Nevertheless, Table 3 also reports runtime, explanation size, and multiple plausibility metrics (IM1, IM2, Implausibility, and Moran’s I), which together provide insight into interpretability.
>
> We fully agree on the importance of human-centred evaluation. While our primary goal was reproducing Kenny and Keane’s findings across architectures, we will:
>
> - Conduct a human evaluation study to assess how Moran’s I and other metrics align with human judgments of plausibility and interpretability.
> - Clarify that Moran’s I is not a universal plausibility metric, but an indicator of structural coherence, supported by empirical and human evidence.
> - Clarify in the abstract (and potentially title) our focus on both validity (“success”) and plausibility.
>
> These additions address Weakness #1, Weakness #3, and Requested Change #3.
>
>
> __Weakness #2 – On evaluating with larger models and datasets__
>
> We agree that including a modern architecture would strengthen relevance. Practical constraints (mixed frameworks and computational limits of PIECE) prevented scaling all experiments. However, to address this concern, we will:
>
> - Add a case study using Vision Transformers (ViTs) on ImageNet to evaluate whether plausible counterfactual explanations, such as PIECE,  remain feasible in this setting.
> - Report feasibility, success rates, runtime, explanation size, and plausibility metrics if explanations can be generated.
>
> This directly addresses Weakness #2 and Requested Change #1.
>
>
> __Weakness #4 and Weakness #5 – On Robustness and the connection with adversarial examples__
>
> We appreciate the suggested literature on adversarial examples. Our comment on “inherent lack of robustness” referred to prior work (Jiang et al., 2023; Fokkema et al., 2023) and was intended as motivation, not an evaluation claim. Robustness metrics (e.g., delta-robustness) are outside the scope of this reproducibility study. Nevertheless, we will:
>
> - Add a dedicated discussion linking counterfactual explanations to adversarial-example literature (e.g., Pawelczyk et al., 2021; Kaur et al., 2019; Santurkar et al., 2019).
> - Remove any implied claims of robustness evaluation.
>
> This addresses Weakness #4, Weakness #5, and Requested Change #4.
>
>
> __Are the claims made supported by accurate, convincing and clear evidence?__
>
> We appreciate the reviewer’s note that some claims were phrased too strongly. The Moran’s I claim is addressed above. Regarding Figure 3, its purpose is to show that method rankings vary significantly across architectures. We focused on rankings to visualise multi-architecture performance more compactly. However, we will:
>
> - Rephrase the discussion to clarify the figure’s intent.
> - Improve clarity around ranking interpretation.
> - Include success metric values in Figure 3 if this improves clarity.
>
>
> __Would TMLR’s audience be interested in this work?__
>
> While we respectfully disagree that TMLR’s audience would not find interest in reproducibility studies, we take the reviewer’s concerns seriously. We will revise the introduction and conclusion to clarify:
>
> - Why reproducibility in counterfactual explanations is necessary.
> - How our work addresses gaps in evaluation practice.
> - How the ViT case study and human evaluation broaden relevance.
>
> We thank the reviewer again for their thoughtful input. We believe the planned human study, ViT case study, clarified use of Moran’s I, and expanded discussion of robustness will substantially strengthen the paper while remaining aligned with its reproducibility focus.

---

### Review · Reviewer_2pwk · 2025-12-02

**Summary Of Contributions:**

This paper states that the comparison of methods for counterfactuals is hindered by the lack of a standardized experimental setup. To address this, it evaluates five methods for visual counterfactual explanations that differ only in the loss function to produce these explanations, various CNN- and MLP-based classifiers (but no Vision Transformer), two datasets (MNIST and CIFAR-10), and eight automatic evaluation metrics (including one proposed by the paper). It also studies the fraction of successful counterfactual explanations across original class and target class pairs, and the agreement of plausibility metrics.

I am not an expert on visual counterfactual explanations and am definitely not up to date with the literature in this space. I don’t feel like I can trust that the paper has done a comprehensive enough literature review to motivate a need for another reproducibility study of the evaluation of methods for visual counterfactual explanations. I provide more concrete explanations below (see Paper Quote 1, Paper Quote 2).

**Additional Comments:**

*  This claim is unsupported: “Through time, models and tasks have become increasingly complex, making the black-box even more opaque.” Interpreting BERT (2018) doesn't seem any easier than interpreting a 2025 LLM like Qwen 3, as both have internal mechanisms obscured in complex systems of high-dimensional vectors. I recommend removing this sentence.

* It would be better not to assume that a reader knows what “recourse sensitive” means, but to briefly explain.

* The logic at the end of the first paragraph seems to be contradictory. If counterfactual explanations “can never be robust”, then the takeaway calling for “rigorous evaluation” seems unnecessary as we already know they are not robust. Why should we rigorously evaluate a property that cannot exist? The author should revise their motivation for robust evaluation of counterfactuals.

**Audience:**

No

**Audience Explanation:**

I'd choose that I can't determine if that was an option. Let me explain; in the process of verifying this:

> **(Paper Quote 2)** “Benchmarks often have different implementations of the same counterfactual explanation algorithm and of the metrics used to evaluate those explanations (Karimi et al., 2022; Le et al., 2023), making it impossible to compare the performance of counterfactual explanation methods.”

I came across Vaeth et al. (PKDD/ECML Workshops 2023) “Diffusion-based Visual Counterfactual Explanations - Towards Systematic Quantitative Evaluation”,  which states:

> “One of the difficulties of research in counterfactual methods is the lack of broadly accepted evaluation procedures. [...] In this paper we contribute to the discussion on best evaluation practices for VCE [visual counterfactual explanations] methods. We review a number of previously proposed metrics and identify the most useful ones for assessing the properties of natural image VCEs (Section 3). Using these, we systematically quantitatively evaluate recent, diffusion-based, generative VCE methods [1, 15] across a range of robust and non-robust classifiers (Section 5). We complement this by an ablation of critical design choices that we identified as promising candidates for future improvements (Section 2.3). We provide access to our streamlined implementation of the VCE methods containing the ablation options as well as additional classifiers not present in the original code.”

The paper under review doesn't cite Vaeth et al. (2023) or position itself against this related work. Since both papers are motivated by the same problem (lack of systematic evaluation for visual counterfactual explanations) and both systematically evaluate VCE methods using automated metrics, I cannot assess whether anyone in the TMLR audience would find the reviewed paper necessary and therefore interesting without understanding how it differs from Vaeth et al. (2023), or any other related work that is motivated by the same problem and also systematically evaluate VCE methods.

**Broader Impact Concerns:**

None.

**Claims And Evidence:**

No

**Claims Explanation:**

The paper states:

> **(Paper Quote 1)** “Within the area of counterfactual explanations, there is also no uniform [...] set of evaluation techniques (Stepin et al., **2021**).”

This statement may be too broad and outdated, as Nguyen et al. (INLG **2024**) “CEval: A Benchmark for Evaluating Counterfactual Text Generation” introduced a benchmark to establish a unified evaluation standard for counterfactuals in NLP.

**Requested Changes:**

**Re: Paper Quote 1**

 The authors should revise to either explain why their statement is true across fields (vision, NLP, tabular data) or specify AI fields for which it is true.

**Re: Paper Quote 2**

* Better position their work against Vaeth et al. (2023) and any other work that also aims to systematically evaluate visual counterfactual explanations. I am in particular concerned about research questions 1 and 2 that analyze methods across metrics and neural architectures, which this paper also seems to do.

* If the authors can present a convincing argument that their work is valuable in addition to Vaeth et al. (2023) and potentially other such papers, it would greatly help to appreciate this reproducibility study if Paper Quote 2 is backed with a table showing which evaluation metrics and datasets are used across published counterfactual papers. Rows could be papers, columns could be metrics and datasets, and cells could indicate whether each metric/dataset is used. The authors could use the Semantic Scholar API to find relevant papers (keywords like “counterfactual” in computer vision venues, 2020–2025), and if too many results come up, analyzing the top-20 by citation count would be sufficient.

**Other**

* Justify why newer methods for counterfactual generations, such as DIME (Jeanneret et al. 2022. Diffusion models for counterfactual explanations) and DVCE (Augustin et al. 2022. Diffusion visual counterfactual explanations) aren’t evaluated given that the stated goal is to analyze “the **current state** of counterfactual explanations for image classifiers”. Similarly, while MNIST and CIFAR-10 may be the most commonly used datasets, a comprehensive comparison should aim for the union of evaluation settings across the literature, not just the intersection. Another timely choice would be to use Vision Transformer, as downstream users of explainability methods want explanations for state-of-the-art classifiers.

---

> ### Author Response · Authors · 2025-12-21
> **Response to Reviewer 2pwk**
>
> We thank the reviewer for the thoughtful and detailed feedback. We appreciate the time taken to identify missing citations and unclear motivations. The suggestions will meaningfully improve both the clarity and positioning of our work. Below, we respond to the reviewer’s comments point-by-point and outline planned revisions.
>
> The reviewer mentions that they do not feel confident that the paper presents a sufficient analysis of prior work to motivate another reproducibility study. While we are grateful for the additional relevant papers suggested, we believe a reproducibility study is justified in this case, as we specifically reproduce the PIECE method by Kenny and Keane (AAAI 2021). We argue that reproducing this method across a wider range of architectures, decision boundaries, and plausibility metrics is valuable, as it helps future researchers assess how applicable plausibility-inspired methods are in different contexts. This is also aligned with TMLR’s scope, which regularly includes reproducibility studies. This level of analysis was not conducted in the original paper.
>
> __Are the claims made in the submission supported by accurate, convincing and clear evidence?__
>
> We thank the reviewer for bringing Nguyen et al. to our attention and will include this work in the revised paper. However, we respectfully disagree that our original statement is outdated. In explainable AI, it remains an open challenge to define what a “good” explanation is, as multiple criteria exist and the domain influences which metrics are appropriate (Guidotti, 2022; Nauta et al., 2023). There is no single evaluation standard that transfers directly across domains. That said, we agree this nuance should be made clearer. We will:
>
> - Add a discussion in the related work section on existing benchmarks in other domains, including Nguyen et al..
> - Refine the quoted sentence to better reflect the domain-specific nature of counterfactual explanation evaluation.
>
>
>
> __Would at least some individuals in TMLR's audience be interested in knowing the findings of this paper?__
>
> We again thank the reviewer for highlighting relevant prior work and agree that it should be cited. However, we believe our work will be of interest to part of the TMLR audience, as it places a stronger emphasis on plausibility metrics and plausibility-driven methods than prior benchmarking work such as Vaeth et al. In the revised paper, we will:
>
> - Include a reference to Vaeth et al. and clearly discuss how our work differs in scope and emphasis.
>
>
> __Addressing requested changes__
>
> > “The authors should revise to either explain why their statement is true across fields (vision, NLP, tabular data) or specify AI fields for which it is true.”
>
> - Our focus is on vision, specifically image classification, as stated in the title. We will revise the manuscript to ensure this scope is clear throughout.
>
> > “Better position their work against Vaeth et al. (2023) and other systematic evaluations.”
>
> - We will strengthen our motivation by explicitly positioning our reproducibility study relative to Vaeth et al. (2023). Additionally, we plan to examine how plausibility metrics align with human judgments, an aspect not addressed by Vaeth et al.
>
> - We will also include a case study under Research Question 2 to assess the feasibility of applying methods such as PIECE to larger models, specifically Vision Transformers, on a dataset like ImageNet.
>
> > “Include a table comparing evaluation metrics and datasets across counterfactual papers.”
>
> - We agree this would strengthen the positioning of our work and will include such a table in the related work section of the revised paper.
>
> > “Justify why newer diffusion-based methods (e.g., DIME, DVCE) are not evaluated.”
>
> - We will clarify that our goal is not to provide an exhaustive benchmark of all existing methods, but to reproduce a specific method and its baselines across varied settings. As noted above, we will nevertheless strengthen Research Question 2 with a case study
>
> > “This claim is unsupported: ‘Through time, models and tasks have become increasingly complex…’”
>
> - We will remove this sentence as suggested.
>
> > “Explain what ‘recourse sensitive’ means.”
>
> - We will expand our discussion of Fokkema et al. and clarify the implications of recourse sensitivity for counterfactual explanations.
>
> > “Clarify the motivation for robustness evaluation.”
>
> - We will revise this section to clarify that although prior theoretical work suggests counterfactual explanations are not robust, robustness is only one dimension of explanation quality. Other properties such as validity, plausibility, and size remain important to evaluate.
>
> We thank the reviewer again for their valuable feedback. We believe these revisions will significantly improve the clarity, rigor, and contextual grounding of the paper while remaining aligned with its scope as a reproducibility study.

---

### Review · Reviewer_TzKm · 2025-12-17

**Summary Of Contributions:**

This paper presents a systematic evaluation of counterfactual explanation methods for image classification. The authors benchmark five existing counterfactual generation techniques across multiple neural network architectures, target classes, and plausibility metrics on MNIST and CIFAR-10.

**Additional Comments:**

1. How do you expect your findings to translate to ImageNet-scale models and datasets?


2. Have you observed cases where Moran’s I is high but explanations are semantically implausible?


3. Can you provide guidance on which plausibility metric practitioners should prefer in practice?

**Audience:**

Yes

**Audience Explanation:**

It provides a systematic empirical analysis of counterfactual explanations and highlights overlooked factors such as architecture and target-class dependence, along with reproducibility insights.

**Broader Impact Concerns:**

No major broader impact concerns are identified.

**Claims And Evidence:**

Yes

**Claims Explanation:**

Yes.
The claims are supported by clear and systematic empirical evidence within the scope of the study. The paper evaluates multiple counterfactual explanation methods across different architectures and target classes on MNIST and CIFAR-10, and the reported results consistently demonstrate substantial performance variation along these dimensions. The experimental design is transparent, failure cases are explicitly reported, and the conclusions directly follow from the presented evidence.

**Requested Changes:**

1. Severely limited experimental scope undermines generality

Despite repeatedly using terms such as “large-scale”, “comprehensive”, and “generalizable”, the experiments are restricted to MNIST and CIFAR-10, both of which are widely regarded as toy or near-toy benchmarks in modern vision research.
No experiments are conducted on ImageNet or any comparable large-scale, semantically rich dataset. As a result, it is unclear whether the reported sensitivities or the proposed Moran’s I diagnostic remain meaningful in realistic vision settings.


2. Questionable relevance of Moran’s I beyond toy settings

Moran’s I measures spatial autocorrelation, which is a reasonable proxy for structural coherence in low-level pixel domains. However:
High Moran’s I does not imply semantic plausibility. Also, the paper does not analyze failure cases where Moran’s I is high but the explanation is visually nonsensical.

---

> ### Author Response · Authors · 2025-12-21
> **Response to Reviewer TzKm**
>
> We thank the reviewer for the positive assessment of the paper’s empirical rigour and relevance, and for the constructive feedback regarding the experimental scope and plausibility metrics. Below, we respond to the requested changes and clarify how we will revise the manuscript.
>
> __On the experimental scope__
>
> We agree that MNIST and CIFAR-10 do not capture the semantic richness of large-scale datasets such as ImageNet. Our goal was not to claim universal generalisability, but to conduct a reproducibility study to isolate the effects of architecture, target class, and the evaluation of explanations using various plausibility metrics. We believe that for the purpose of reproducing the paper *“On Generating Plausible Counterfactual and Semi-Factual Explanations for Deep Learning”* by Kenny and Keane (AAAI 2021), evaluation on these two datasets suffices to get our main points across. Another (practical) reason for focusing on MNIST and CIFAR-10 is that the evaluated counterfactual explanation methods span both PyTorch- and Keras-based implementations. Supporting ImageNet-scale models consistently across frameworks would have required substantial re-engineering and computational costs.
>
> That being said, we acknowledge that our use of terms such as “large-scale” and “generalisable” can be misinterpreted as claims of ImageNet-level coverage. We will revise the language to more precisely reflect the scope of the study and avoid overstatement. Importantly, in response to this and other reviews, we will:
>
> - Add a dedicated case study on ImageNet using a Vision Transformer (ViT). This experiment is explicitly designed to test the extent to which these plausible counterfactual explanations, such as PIECE, generalise to larger datasets and models with a large number of classes.
> - Change the language of our claims to not include terms such as “large-scale” and “generalisable”.
>
> While not intended as an exhaustive ImageNet/ViT benchmark, this addition directly addresses the reviewer’s concern about realism and translation beyond toy datasets.
>
>
> __On the relevance of Moran’s I__
>
> As the reviewer correctly points out, Moran’s I can be a reasonable proxy for structural coherence. However, since semantic plausibility can ultimately only be assessed by humans, we cannot know for certain whether a high Moran’s I score guarantees semantic plausibility. Therefore, we agree with the reviewer that an analysis of the extent to which failure cases occur with Moran’s I would strengthen the work. To address this, we will:
>
> - Add a human evaluation study to assess how Moran’s I and other plausibility metrics align with human judgments; this study will systematically identify cases where metric scores and human perception diverge.
> - Include qualitative examples derived from this study where Moran’s I is high, but the counterfactual is not judged semantically plausible.
>
>
> __Addressing the additional comments__
>
> We believe that the above-mentioned changes will address these comments by:
>
> - Including a case study that shows how plausible counterfactual explanation methods like PIECE hold up to larger models like ViTs and datasets like ImageNet, and what the challenges and implications may be for future practitioners who want to use these explanation methods with these neural architectures.
>
> So far, we have not observed cases where Moran’s I is high, but explanations are semantically implausible. However, we believe the addition of the human evaluation study would systematically answer this question.
>
> Based on the human study, we could give a recommendation on which plausibility metric aligns most with human-perceived semantic plausibility. Currently, based on our experiments, none of the methods agree; therefore, we cannot say which is objectively better. However, IM1 and IM2 require autoencoders trained on each class in the dataset. I would imagine for ImageNet, this would be computationally expensive. Therefore, (im)plausibility and Moran’s I would work better for larger datasets. We will include a discussion on this in the revised paper.
>
> In summary, we appreciate the reviewer’s concerns and believe the planned revisions substantially strengthen the paper. The added ImageNet case study, human plausibility evaluation and clarified claims will directly address the issues raised while preserving the paper’s core contribution: a reproducibility study of plausible counterfactual explanations.

---

### Decision · Action_Editor_3WCJ · 2026-02-05

**Recommendation:** Reject

**Additional Comments:**

I believe that while the authors have given their comments, they have not changed the manuscript. This only strengthens my decision to put the recommendation as a reject, however if the authors wish to do so, they may take into account the comments and resubmit a major revision.

**Audience:**

No

**Audience Explanation:**

There are a number of concerns, voiced by the reviewers, that stops me from answering yes to the question of audience:
- The evaluation is limited to MNIST and CIFAR-10, with limited exploration of state-of-the-art architectures such as Vision Transformers, with no larger-scale experiments. This limits the use of this work for the contemporary computer vision practice.

- There are also concerns about the the use Moran's index, as the authors do not corroborate on the connection between the index and human-perceived plausibility, which could be also of interest to the audience

- Another missing aspect which could ensure the audience is distinguishing this analysis from prior evaluations of visual counterfactual explanations

- Unfortunately this paper does not contain a discussion of any downstream utility of counterfactual explanations, making it unlikely to be of broader interest.

- While it is not a concern of novelty, there should be a clear distinction and additional angle to existing literature. Specifically, the reviewers mention Väth et al. (2023). The authors need to contrast this work with the proposed one, since essentially I understand that both papers have a very similar motivation, precisely lack of systematic evaluation for visual counterfactual explanations, and both systematically evaluate visual counterfactual explanation methods using automated metrics. In other words, if one paper is doing so, the audience needs to know how the angle of this paper complements the angle of the Väth et al (2023) amongst possibly others.
- There was also a point from the reviewers that the authors suggest that the reproducibility of the PIECE method by Kenny and Keane (AAAI 2021) is TMLR-acceptable, despite the existence of newer diffusion-based methods. The concern is that the authors need to justify the reasons why the audience would be interested in the insights about an older method in light of a newer one, which is assumed to be better performing. Therefore, there is a missing part in the argument, that needs to be fleshed out.

In my understanding, none of these are unsolvable, but the aspects mentioned above are not solved yet in the current version of the paper.

--

Väth, Philipp, et al. "Diffusion-Based Visual Counterfactual Explanations-Towards Systematic Quantitative Evaluation." Joint European Conference on Machine Learning and Knowledge Discovery in Databases. Cham: Springer Nature Switzerland, 2023.

**Claims And Evidence:**

No

**Claims Explanation:**

The reviewers mentioned both reasons for and against the statement that the claims are supported by accurate, convincing and clear evidence.

Arguments for it:
- the empirical evidence is clear, systematic and matches the scope of the study, involving multiple counterfactual explanation methods on MNIST and CIFAR-10, with transparent experimental design and explicit reporting of failure cases, with conclusions following from the evidence.

However, there are also concerns about some inaccurate claims:
- the claim that Moran's index can be used as a measure of explanation quality: this claim should be substantiated (see also the audience criterion)
- it is claimed that the performance can vary depending up the network architecture. However, instead of directly measuring it and presenting success metrics directly, it ranks the model performance. The model performance can fluctuate, but it does not necessarily imply that the difference is statistically significant (it may as well, but the authors need to show it).

In the discussion, the authors have stated how they could improve upon the current claims and make the support for the claims stronger. For the paper to meet the claims and evidence criterion, it has to be done.

**Resubmission Of Major Revision:**

The authors may consider submitting a major revision at a later time.